# IDO1$^+$ Paneth cells promote immune escape of colorectal cancer

Sandra Pflügler[1,17], Jasmin Svinka[1,17], Irene Scharf[1], Ilija Crncec[1], Martin Filipits [1], Pornpimol Charoentong[2,3,4], Markus Tschurtschenthaler [5,6], Lukas Kenner[7,8,9], Monira Awad[1], Judith Stift[8], Marina Schernthanner[1], Romana Bischl[1], Dietmar Herndler-Brandstetter[1], Elisabeth Glitzner[1], Herwig P. Moll [10], Emilio Casanova [7,10], Gerald Timelthaler[1], Maria Sibilia[1], Michael Gnant [11], Sigurd Lax[12,13], Josef Thaler[14], Mathias Müller [15], Birgit Strobl [15], Thomas Mohr [1], Arthur Kaser[5], Zlatko Trajanoski [2], Gerwin Heller[16] & Robert Eferl [1✉]

Tumors have evolved mechanisms to escape anti-tumor immunosurveillance. They limit humoral and cellular immune activities in the stroma and render tumors resistant to immunotherapy. Sensitizing tumor cells to immune attack is an important strategy to revert immunosuppression. However, the underlying mechanisms of immune escape are still poorly understood. Here we discover Indoleamine-2,3-dioxygenase-1 (IDO1)$^+$ Paneth cells in the stem cell niche of intestinal crypts and tumors, which promoted immune escape of colorectal cancer (CRC). Ido1 expression in Paneth cells was strictly Stat1 dependent. Loss of IDO1$^+$ Paneth cells in murine intestinal adenomas with tumor cell-specific *Stat1* deletion had profound effects on the intratumoral immune cell composition. Patient samples and TCGA expression data suggested corresponding cells in human colorectal tumors. Thus, our data uncovered an immune escape mechanism of CRC and identify IDO1$^+$ Paneth cells as a target for immunotherapy.

[1] Institute of Cancer Research, Medical University of Vienna and Comprehensive Cancer Center, 1090 Vienna, Austria. [2] Institute of Bioinformatics, Medical University Innsbruck, Biocenter, 6020 Innsbruck, Austria. [3] Department of Medical Oncology, National Center for Tumor diseases, University Hospital Heidelberg, 69120 Heidelberg, Germany. [4] German Cancer Research Center (DKFZ), 69120 Heidelberg, Germany. [5] Division of Gastroenterology and Hepatology, Department of Medicine, University of Cambridge, Cambridge CB2 0QQ, UK. [6] Center for Translational Cancer Research (TranslaTUM), Technical University of Munich, 81675 Munich, Germany. [7] Ludwig Boltzmann Institute for Cancer Research LBICR, 1090 Vienna, Austria. [8] Institute of Clinical Pathology, Medical University of Vienna, 1090 Vienna, Austria. [9] Department of Laboratory Animal Pathology, University of Veterinary Medicine Vienna, 1210 Vienna, Austria. [10] Department of Physiology, Center of Physiology and Pharmacology, Comprehensive Cancer Center (CCC), Medical University of Vienna, 1090 Vienna, Austria. [11] Department of Surgery, Breast Health Center, Comprehensive Cancer Center, Medical University of Vienna, 1090 Vienna, Austria. [12] Department of Pathology, Hospital Graz II, 8020 Graz, Austria. [13] Institute of Pathology and Molecular Pathology, Johannes Kepler University, 4040 Linz, Austria. [14] Department of Internal Medicine IV, Klinikum Wels-Grieskirchen, 4600 Wels, Austria. [15] Institute of Animal Breeding and Genetics, University of Veterinary Medicine Vienna, 1210 Vienna, Austria. [16] Division of Oncology, Medical University of Vienna and Comprehensive Cancer Center, 1090 Vienna, Austria. [17] These authors contributed equally: Sandra Pflügler, Jasmin Svinka. ✉email: robert.eferl@meduniwien.ac.at

Colorectal cancer (CRC) is the third most common cancer worldwide and patients with metastases in distant organs have a 5-year survival rate below 13%[1]. Metastatic CRC is currently treated with several combinations of cytotoxic agents. They improved overall survival of patients, treated initially with fluoropyrimidine monotherapy, from 12 to 30 months[2]. However, chemotherapy reached its limits[3], which fostered clinical trials for immunotherapies[4]. The importance of immuno-surveillance in CRC is emphasized by the good prognostic value of CD3[+], CD8[+], and CD45RO[+] T-cell infiltration (Immuno-score)[5–7]. Unfortunately, immunotherapy with checkpoint inhibitors showed clinical benefits only in mismatch-repair-deficient CRC with high neo-antigen load. Durable responses in CRC with different etiologies remained scarce, which is due to immune escape mechanisms[8].

The transcription factor signal transducer and activator of transcription 1 (Stat1) is a key effector in tumor immuno-surveillance mediated by natural killer (NK)- and T cells[9,10]. Consistently, Stat1 is part of the immunologic constant of rejection gene expression signature, which correlates with good prognosis of CRC[7]. In cancer cells, Stat1 inhibits proliferation and promotes apoptosis via induction of cyclin-dependent kinase inhibitors and pro-apoptotic proteins[9]. Stat1 also regulates the expression of tissue antigens and proteins of the antigen pre-sentation machinery, which enhance the immunogenicity of tumors[11]. Therefore, it is generally considered that Stat1 expression and activation in immune cells and in cancer cells suppresses tumorigenesis. Type I and II interferon (IFN) are the major activators of canonical Stat1 signaling, which relies on Tyr701 phosphorylation (pY-STAT1) and mediates tumor sup-pressive effects of IFN[12]. However, the tumor cell-intrinsic role of Stat1 in CRC is not well defined.

Lgr5[+] stem cells at the bottom of intestinal crypts have been identified as possible precursor cells for CRC[13]. However, non-stem cells can also acquire tumor-initiating capacity[14] and Lgr5[+] cancer stem cells are not essential for growth of primary tumors[15]. These cells are separated by Lysozyme[+] Paneth cells in the small intestine, which provide essential niche factors for stem cell proliferation and self-renewal[16]. In the colon, Lysozyme[−] deep crypt base secretory cells support Lgr5[+] stem cells[16] but colonic Paneth cells can appear through epithelial metaplasia[17]. Paneth cells have been identified in intestinal tumors of $Apc^{Min}$ mice[18] and in sporadic CRC of humans, albeit at varying fre-quencies ranging from 0.2% to 39%[19]. However, intestinal tumors from familial adenomatous polyposis (FAP) patients with inherited $Apc$ mutations harbored more than 90% of Paneth cells[20]. The role of Paneth cells is unclear but CRC developed predominantly in colonic mucosal tissue with Paneth cell meta-plasia[21] and the presence of Paneth cell-containing adenomas increased the risk for synchronous CRC[19]. Therefore, Paneth cells might promote CRC formation.

Here we identified a subset of Paneth cells that displayed Stat1-dependent expression of the immune checkpoint molecule IDO1. Loss of these cells in Stat1-deficient intestinal tumors of $Apc^{Min}$ mice resulted in reduced tumor load and increased infiltration of anti-tumor immune cells.

## Results

**Epithelial Stat1 promotes formation of intestinal tumors.** We used $Apc^{Min}$ mice[22] with conditional deletion of $Stat1$ in intest-inal epithelial cells ($Stat1^{\Delta IEC}$ $Apc^{Min}$)[23,24] to identify Stat1 functions in intestinal tumorigenesis. Deletion of $Stat1$ was con-firmed by PCR (Supplementary Fig. 1a), quantitative PCR (qPCR) of purified intestinal epithelial cells (Supplementary Fig. 1b) and immunohistochemistry (IHC, Supplementary

Fig. 1c). Lamina propria cells of $Stat1^{\Delta IEC}$ $Apc^{Min}$ mice displayed STAT1 expression, which demonstrated specific ablation in intestinal epithelial cells (Supplementary Fig. 1c). Goblet, enter-oendocrine, Paneth, and proliferating cells in the intestinal crypts were present at normal numbers in $Stat1^{\Delta IEC}$ $Apc^{Min}$ mice (Supplementary Fig. 1d-h). These data show that epithelial cell-specific deletion of $Stat1$ does not affect intestinal cell differ-entiation and crypt proliferation of $Apc^{Min}$ mice.

Four-month-old $Stat1^{flox/flox}$ $Apc^{Min}$ and $Stat1^{\Delta IEC}$ $Apc^{Min}$ mice were used to investigate epithelial cell-intrinsic functions of Stat1 in intestinal tumorigenesis. Tumor formation was reduced in $Stat1^{\Delta IEC}$ $Apc^{Min}$ male and female mice (Fig. 1a–c). Angiogen-esis (Supplementary Fig. 2a), tumor cell proliferation, and apoptosis (Supplementary Fig. 2b) were not affected but numbers of low-grade adenomas were increased (Fig. 1d). These data show that epithelial cell-intrinsic Stat1 promotes the formation and progression of intestinal tumors in $Apc^{Min}$ mice.

$Apc^{Min}$ mice develop tumors mainly in the small intestine but also in the colon[25]. Similarly, we found tumors in the small intestine and the colon and stained them for STAT1 by IHC. STAT1 was detected in tumor and stroma cells of $Stat1^{flox/flox}$ $Apc^{Min}$ mice. In contrast, STAT1 was not detectable in tumor cells of $Stat1^{\Delta IEC}$ $Apc^{Min}$ tumors demonstrating efficient condi-tional deletion (Fig. 1e–g). However, $Stat1^{\Delta IEC}$ $Apc^{Min}$ tumors displayed a significant upregulation of STAT1 in the tumor stroma (Fig. 1e, g). Numbers of STAT3- and activated pY-STAT3-positive cells were not changed in $Stat1^{\Delta IEC}$ $Apc^{Min}$ tumors (Supplementary Fig. 2c-e). These data show that tumor cell-intrinsic Stat1 suppresses upregulation of Stat1 in the stroma of $Apc^{Min}$ tumors.

**Tumor cell-intrinsic Stat1 suppresses immune cell activation.** We have recently shown that enhanced anti-tumor immune cell activity is reflected by increased stromal Stat1 expression in azoxymethane-dextran sodium sulfate (AOM-DSS)-induced col-orectal tumors[26]. Therefore, we investigated immune cell activa-tion in $Stat1^{\Delta IEC}$ $Apc^{Min}$ tumors. As human intestinal tumors develop mainly in the colon, we performed RNA sequencing (RNA-seq) experiments with murine colon tumors (Supplemen-tary Data 1 and 2). The analyses showed reduced expression of several IFN-stimulated genes (ISGs) in $Stat1^{\Delta IEC}$ $Apc^{Min}$ tumors (Supplementary Data 1). In particular, 24 out of 49 orthologs of human genes of the IFN-related gene signature for DNA damage (IRDS)[27] were downregulated in $Stat1^{\Delta IEC}$ $Apc^{Min}$ tumors (Fig. 1i–j and Supplementary Data 1). We analyzed The Cancer Genome Atlas (TCGA) data to evaluate whether these genes are also regulated by Stat1 in human CRC. A correlation analysis revealed 529 genes that are co-expressed with Stat1 in human CRC (Supplementary Table 1). Among them were 16 IRDS genes. A Venn diagram using (i) human IRDS genes[27], (ii) orthologs of human IRDS genes, significantly downregulated in $Stat1^{\Delta IEC}$ $Apc^{Min}$ mouse tumors, and (iii) the 529 genes, identified to be co-expressed with Stat1 in the CRC TCGA dataset, revealed a sub-stantial overlap. A signature of 13 genes was present in all three gene sets (Fig. 1h). Gene Ontology (GO) term-enrichment ana-lyses of deregulated genes in $Stat1^{\Delta IEC}$ $Apc^{Min}$ tumors and genes that are co-expressed with Stat1 in human CRC showed a sub-stantial overlap of GO terms and revealed mainly pathways implicated in immunological processes (Supplementary Data 3 and 4). These data suggest that Stat1 target genes and Stat1-dependent regulation of immunological processes are similar in human and murine CRC.

Among the most significantly downregulated modulators of immune responses in $Stat1^{\Delta IEC}$ $Apc^{Min}$ tumors was the enzyme Indoleamine-2,3-dioxygenase-1 (Ido1, Fig. 1j, Supplementary

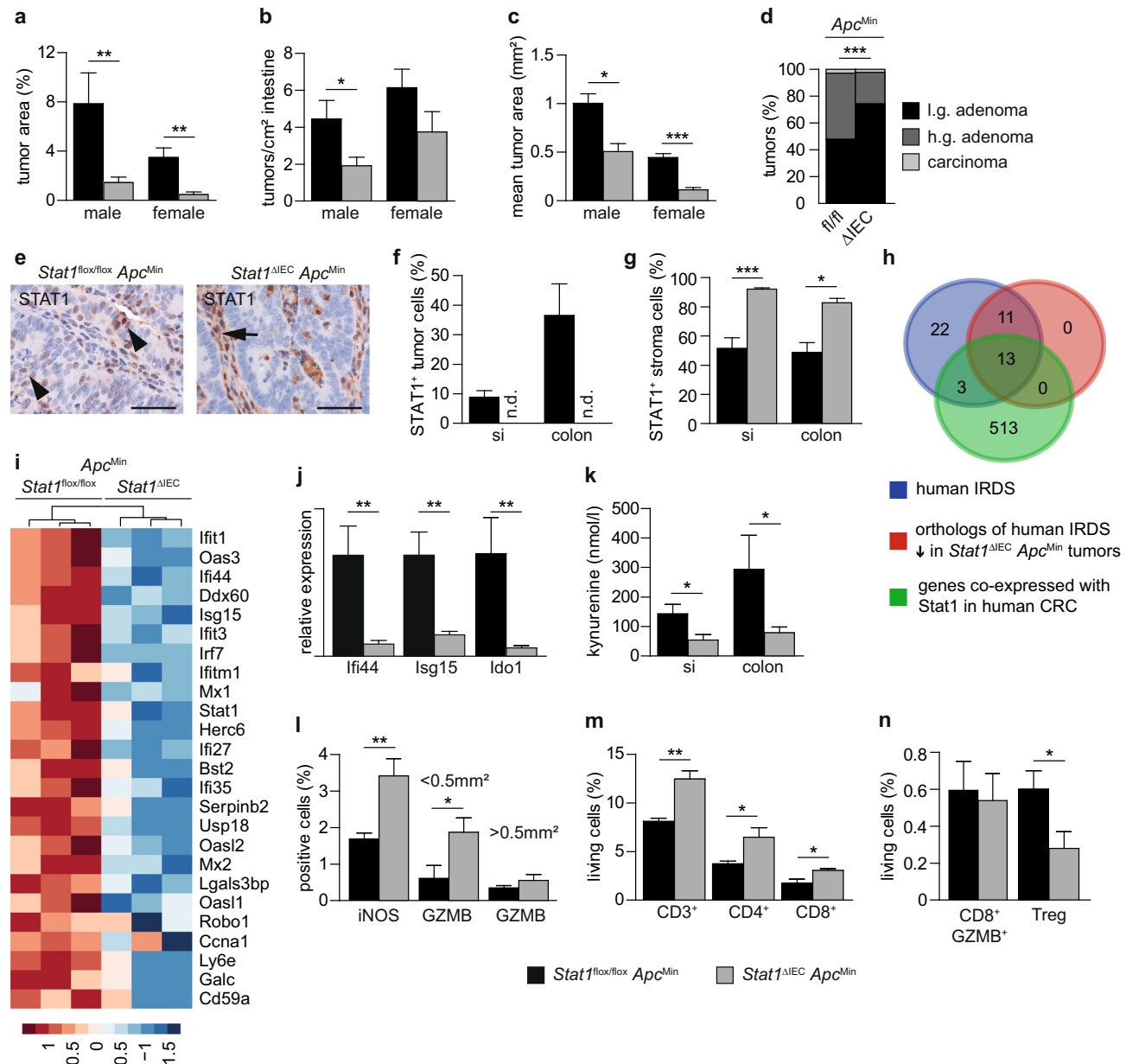

**Fig. 1 Reduced intestinal tumor burden in *Stat1*^ΔIEC *Apc*^Min mice. a–c** Quantification of tumor load (**a**), number (**b**), and mean area (**c**) in *Stat1*^flox/flox *Apc*^Min (7♀, 9♂) and *Stat1*^ΔIEC *Apc*^Min (6♀, 13♂) mice. **d** Grading of tumors in *Stat1*^flox/flox *Apc*^Min (115 tumors, 9 mice) and *Stat1*^ΔIEC *Apc*^Min (68 tumors, 9 mice) mice. Low grade: $p < 0.001$; high grade: $p < 0.001$; carcinoma: $p = 0.473$. **e** IHC staining demonstrating STAT1^+ tumor cells (arrowheads) in *Stat1*^flox/flox *Apc*^Min colon tumors. STAT1 expression was not detectable in tumor cells of *Stat1*^ΔIEC *Apc*^Min tumors but was increased in the stroma (arrow). Scale bars indicate 50 μm. **f, g** Quantification of STAT1^+ tumor (**f**) and stroma (**g**) cells (**f**: *Stat1*^flox/flox *Apc*^Min si 12 tumors, 4 mice; colon 13 tumors, 4 mice; *Stat1*^ΔIEC *Apc*^Min si 12 tumors, 4 mice; colon 12 tumors, 4 mice; **g**: 24 tumors of 4 mice each). **h** Venn diagram showing an overlap between human IRDS genes, IRDS genes downregulated in *Stat1*^ΔIEC *Apc*^Min colon tumors, and genes that positively correlated with Stat1 expression in human CRC TCGA data. **i** Heat map of downregulated IRDS genes in *Stat1*^ΔIEC *Apc*^Min colon tumors. **j** qPCR for Ifi44, Isg15, and Ido1 mRNA expression in colon tumors of *Stat1*^flox/flox *Apc*^Min (5 mice) and *Stat1*^ΔIEC *Apc*^Min (6 mice) mice (tumors from each mouse were pooled). **k** ELISA for kynurenine in supernatants of *Stat1*^flox/flox *Apc*^Min (si: 7 tumors, 5 mice; colon: 7 tumors, 6 mice) and *Stat1*^ΔIEC *Apc*^Min tumors (si: 7 tumors, 7 mice; colon: 7 tumors, 5 mice). **l** Quantification of iNOS^+ and Granzyme B^+ stroma cells (*Stat1*^flox/flox *Apc*^Min iNOS: 12 tumors, 4 mice; GZMB < 0.5 mm²: 9 tumors, 2 mice; >0.5 mm²: 21 tumors, 4 mice; *Stat1*^ΔIEC *Apc*^Min iNOS: 14 tumors, 4 mice; GZMB < 0.5 mm²: 23 tumors, 5 mice; >0.5 mm²: 13 tumors, 3 mice). **m, n** FACS analysis of immune cells of *Stat1*^flox/flox *Apc*^Min (pooled tumors of 5 mice in 4 experiments) and *Stat1*^ΔIEC *Apc*^Min (pooled tumors of 4 mice in 4 experiments) colon tumors. si: small intestine. Bars represent mean ± SEM.

Fig. 3a, and Supplementary Data 1), which acts as an immune checkpoint[28]. Other immune checkpoints were not deregulated in *Stat1*^ΔIEC *Apc*^Min tumors but expression of the T-cell activation marker CD28 was increased (Supplementary Fig. 3a). IDO1 converts tryptophan into the immune-suppressive metabolite

kynurenine. Consistently, levels of kynurenine were reduced in supernatants of *Stat1*^ΔIEC *Apc*^Min tumors (Fig. 1k). We analyzed expression of inducible nitric oxide synthase (iNOS) and the serine protease Granzyme B, because they are markers for activation of several immune cells such as macrophages, mature

dendritic cells, cytotoxic T cells, or NK cells. IHC characterization of the stroma showed increased numbers of activated iNOS$^+$ immune cells (Fig. 1l). Numbers of Granzyme B$^+$ cells were not significantly changed in large tumors but accumulated in small adenomas of $Stat1^{\Delta IEC}$ $Apc^{Min}$ mice (Fig. 1l and Supplementary Fig. 3b). Fluorescence-activated cell sorting (FACS) analysis of T cells demonstrated increased numbers of CD3$^+$, CD4$^+$, and CD8$^+$ immune cells (Fig. 1m). However, the relative percentage of Granzyme B$^+$ cells among the CD8$^+$ population was not changed (Fig. 1n and Supplementary Fig. 3c). As kynurenine promotes differentiation of regulatory T cells (Tregs)[29], we analyzed CD4$^+$ CD25$^+$ FoxP3$^+$ cell infiltration by FACS. Treg numbers were significantly reduced in $Stat1^{\Delta IEC}$ $Apc^{Min}$ tumors (Fig. 1n and Supplementary Fig. 3c). It has also been reported that kynurenine promotes β-catenin nuclear localization in intestinal cancer cells[30,31] but IHC staining revealed unchanged nuclear β-catenin levels in $Stat1^{\Delta IEC}$ $Apc^{Min}$ tumor cells (Supplementary Fig. 3d). These data suggest that tumor cell-intrinsic Stat1 suppresses stroma immune cell activation in $Apc^{Min}$ tumors through Ido1.

**Stat1 promotes Ido1 expression in neoplastic Paneth cells.** We performed IHC and IF staining to assess downregulation of Ido1 in $Stat1^{\Delta IEC}$ $Apc^{Min}$ tumors at the cellular level. This analysis revealed specific IDO1$^+$ tumor cells in $Stat1^{flox/flox}$ $Apc^{Min}$ tumors that were absent in $Stat1^{\Delta IEC}$ $Apc^{Min}$ tumors (Fig. 2a, b, i). It has been shown that commercially available IDO1 antibodies are unspecific and unable to detect IDO1 in western blottings[32]. To address this issue for IHC staining, we performed in-situ hybridization (ISH) experiments and verified loss of IDO1$^+$ tumor cells at the RNA level (Fig. 2a). The IDO1$^+$ tumor cells were arranged in an alternating pattern with IDO1$^-$ tumor cells in neoplastic adenoma sheets (Fig. 2a). A similar arrangement was described for transformed Lgr5$^+$ stem cells and Lysozyme$^+$ Paneth cells in adenoma sheets of $Apc^{Min}$ tumors, which resembles crypt organization of the small intestine[18]. This suggests that IDO1$^+$ tumor cells are either related to stem cells or to Paneth cells. Co-expression of Lgr5 and Ido1 mRNA was barely detectable by ISH in $Stat1^{flox/flox}$ $Apc^{Min}$ tumor cells (Fig. 2a), indicating that IDO1$^+$ cells are Paneth cells. However, the ISH signals for Lgr5 were weak and not clearly attributable to individual cells. Therefore, double immunofluorescence (IF) with Paneth cell markers was performed. These analyses revealed protein expression of Paneth cell markers Lysozyme and MMP7 in IDO1$^+$ tumor cells (Fig. 2c, g, h). More than 80% of IDO1$^+$ tumor cells expressed Paneth markers (Fig. 2e, k), indicating that Paneth cells are the major source for Ido1 expression in the neoplastic epithelium. Double-positive cells were absent in $Stat1^{\Delta IEC}$ $Apc^{Min}$ tumors (Fig. 2c, g) but the overall numbers of Lysozyme$^+$ and MMP7$^+$ Paneth cells were not reduced (Fig. 2d, j). Moreover, about 50% of Paneth cells expressed IDO1 in $Stat1^{flox/flox}$ $Apc^{Min}$ tumors (Fig. 2f, l). These data demonstrate that Stat1 is required for the formation of IDO1$^+$ Paneth cells in $Apc^{Min}$ tumors. IDO1$^+$ Paneth cells are potential immunosuppressors and their absence might account for immunological changes in $Stat1^{\Delta IEC}$ $Apc^{Min}$ tumors. This assumption is challenged by the low number of IDO1$^+$ Paneth cells in $Stat1^{flox/flox}$ $Apc^{Min}$ tumors (Fig. 2b, i). However, we identified a much higher percentage of IDO1$^+$ Paneth cells in small and early $Apc^{Min}$ adenomas with almost 30% in the colon (Fig. 2m, n). These data suggest that IDO1$^+$ Paneth cells support immune escape during early stages of $Apc^{Min}$-induced tumorigenesis.

Ido1 is an ISG in human and murine tumor cells[33], indicating a role of canonical Stat1 signaling in the formation of IDO1$^+$ Paneth cells. IHC staining for canonical STAT1 activation

detected <3% pY-STAT1$^+$ tumor cells in $Stat1^{flox/flox}$ $Apc^{Min}$ tumors (Supplementary Fig. 4a, b). The pY-STAT1$^+$ tumor cells appeared as cell clusters (Supplementary Fig. 4a), which differed from the alternating arrangement of IDO1$^+$ Paneth cells. pY-STAT1 was undetectable in tumor cells of $Stat1^{\Delta IEC}$ $Apc^{Min}$ tumors (Supplementary Fig. 4a, b) but appeared upregulated in the stroma (Supplementary Fig. 4a, c) similar to upregulation of total STAT1 (Fig. 1e, g). We blunted type I IFN signaling by conditional deletion of $Ifnar1^{34}$ in $Apc^{Min}$ tumor cells. Tumor formation was not affected in the intestine of $Ifnar1^{\Delta IEC}$ $Apc^{Min}$ mice (Supplementary Fig. 4d-f). Moreover, the number of IDO1$^+$ Paneth cells was comparable in $Ifnar1^{flox/flox}$ $Apc^{Min}$ and $Ifnar1^{\Delta IEC}$ $Apc^{Min}$ intestinal tumors (Supplementary Fig. 4g). These data demonstrate that the formation of IDO1$^+$ Paneth cells in tumors of $Apc^{Min}$ mice is independent of type I IFN signaling.

We next investigated whether human CRC contain IDO1$^+$ Paneth cells. Biopsies of 149 human T3 and T4 CRC that had not yet metastasized (Supplementary Table 2) were IHC-stained for STAT1 and IDO1 to compare staining patterns with $Stat1^{flox/flox}$ $Apc^{Min}$ mouse tumors. STAT1$^+$ and IDO1$^+$ cancer cells were readily detectable in human CRC but unlike mouse tumors, IDO1$^+$ cells did not show an alternating pattern with IDO1$^-$ cells (Fig. 3a). However, a TCGA-based correlation matrix of ISGs (IRDS genes) and marker genes for cell identities revealed a correlation between the expression of Stat1, Ido1, and Lysozyme. Lysozyme clustered with Ido1- and Stat1-regulated IRDS genes (Fig. 3b). Moreover, we stained 14 early adenomas (5 from FAP patients) for neoplastic Lysozyme$^+$ IDO1$^+$ Paneth cells. Lysozyme$^+$ and Lysozyme$^+$ IDO1$^+$ Paneth cells were found in ten adenomas (four from FAP patients) and six adenomas (three from FAP patients), respectively. In particular, FAP adenomas showed a perinuclear signal for IDO1 in Paneth cells (Fig. 3c). The relative contribution of Paneth cells to Ido1 expression was assessed by IF staining of the six adenomas harboring IDO1$^+$ Paneth cells. About 50% of IDO1$^+$ tumor cells were Lysozyme-positive, demonstrating a significant contribution of Paneth cells to Ido1 expression in the neoplastic epithelium (Fig. 3d). In summary, these data suggest that IDO1$^+$ Paneth cells are present in human CRC.

**IDO1$^+$ CRC cells promote immune escape.** Subcutaneous implantation of C57BL/6-derived MC38 cells into immunocompetent host mice is an established method for evaluation of pre-clinical immunotherapy approaches[35]. We transplanted green fluorescence protein (GFP)-labeled MC38 CRC cells to test whether deletion of Ido1 in neoplastic cells mimics immunologic consequences of IDO1$^+$ Paneth cell ablation in $Stat1^{\Delta IEC}$ $Apc^{Min}$ tumors. Two independent MC38$^{\Delta Ido1-GFP}$ subclones (MC38$^{\Delta Ido1-GFP-2}$ and MC38$^{\Delta Ido1-GFP-6}$) with CRISPR/Cas9-mediated deletion of the Ido1 locus were generated. The presence of INDELs in MC38$^{\Delta Ido1-GFP}$ cells was verified by sequence analysis. Both clones contained an additional G in exon 6 of Ido1, which is a common CRISPR/Cas9-mediated insertion, and resulted in a truncated IDO1 protein (MC38$^{\Delta Ido1-GFP-2}$ cells are shown in Supplementary Fig. 5a, b). Sequencing revealed also a bigger deletion of the genomic locus upstream of the sgRNA targeting site in MC38$^{\Delta Ido1-GFP-2}$ cells, which might destabilize the Ido1 mRNA. Commercial antibodies are not suitable to detect a specific IDO1 protein by western blotting[32]. However, IFNγ stimulation induced Ido1 mRNA expression in MC38$^{wt-GFP}$ cells but not in MC38$^{\Delta Ido1-GFP-2}$ or MC38$^{\Delta Ido1-GFP-6}$ cells (Fig. 4a and Supplementary Fig. 5c). MC38$^{\Delta Ido1-GFP}$ cells displayed a reduced cumulative cell number (Supplementary Fig. 5d), indicating an in-vitro proliferation defect similar to human cells[31]. In vivo, MC38$^{\Delta Ido1-GFP}$ cells formed smaller tumors than MC38$^{wt-GFP}$

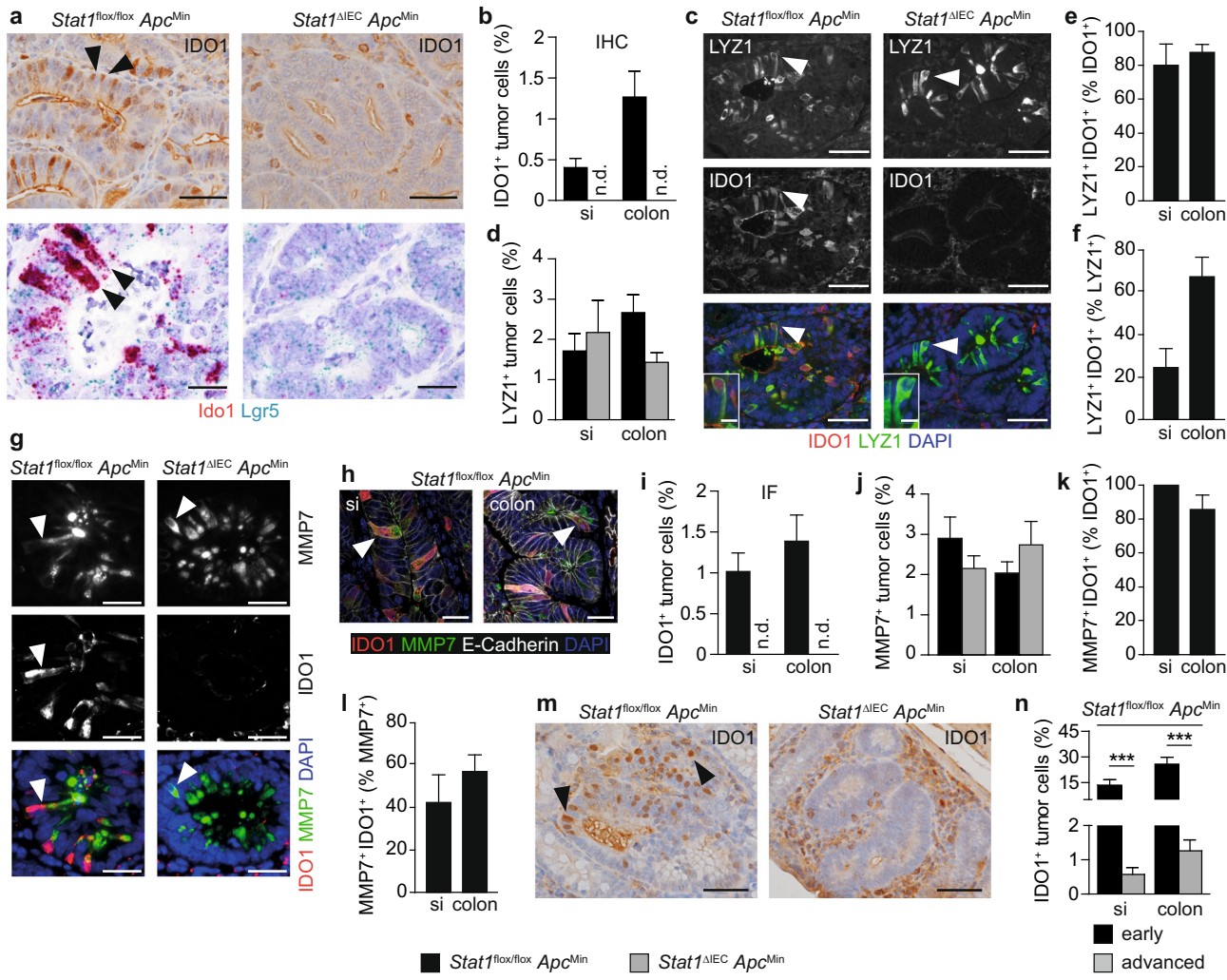

**Fig. 2 Intestinal tumors of *Stat1*<sup>ΔIEC</sup> *Apc*<sup>Min</sup> mice lack IDO1<sup>+</sup> Paneth cells. a** IHC staining for IDO1 (upper panel) and double ISH for Ido1 and Lgr5 mRNA (lower panel) in *Stat1*<sup>flox/flox</sup> *Apc*<sup>Min</sup> and *Stat1*<sup>ΔIEC</sup> *Apc*<sup>Min</sup> colon tumors. IDO1<sup>+</sup> cells are indicated by arrowheads. Scale bars indicate 20 μm.
**b** Quantification of IDO1<sup>+</sup> tumor cells in IHC-stained tumors of *Stat1*<sup>flox/flox</sup> *Apc*<sup>Min</sup> (si and colon: 13 tumors, 4 mice) and *Stat1*<sup>ΔIEC</sup> *Apc*<sup>Min</sup> (si: 12 tumors, 5 mice; colon: 13 tumors, 4 mice) mice. **c** IF staining for IDO1 and Lysozyme (LYZ1) in *Stat1*<sup>flox/flox</sup> *Apc*<sup>Min</sup> and *Stat1*<sup>ΔIEC</sup> *Apc*<sup>Min</sup> colon tumors. Single fluorescent channels and composites are shown. Positive or double-positive cells are indicated by arrowheads. Scale bars indicate 50 or 10 μm (high-magnification insets). **d** Quantification of LYZ1<sup>+</sup> tumor cells of *Stat1*<sup>flox/flox</sup> *Apc*<sup>Min</sup> (si and colon: 12 tumors, 4 mice) and *Stat1*<sup>ΔIEC</sup> *Apc*<sup>Min</sup> (si: 8 tumors, colon 12 tumors, 4 mice each) tumors. **e, f** Quantification of percentages of LYZ1<sup>+</sup> IDO1<sup>+</sup> tumor cells between IDO1<sup>+</sup> (**e**) and LYZ1<sup>+</sup> (**f**) tumor cells of *Stat1*<sup>flox/flox</sup> *Apc*<sup>Min</sup> tumors. **g** IF staining for IDO1 and MMP7 in *Stat1*<sup>flox/flox</sup> *Apc*<sup>Min</sup> and *Stat1*<sup>ΔIEC</sup> *Apc*<sup>Min</sup> colon tumors. Single fluorescent channels and composites are shown. Positive or double-positive cells are indicated by arrowheads. Scale bars indicate 50 μm. **h** IF staining for IDO1, MMP7, and E-Cadherin in *Stat1*<sup>flox/flox</sup> *Apc*<sup>Min</sup> tumors of the small intestine and colon. Triple-positive cells are indicated by arrowheads. Scale bars indicate 50 μm.
**i, j** Quantification of IDO1<sup>+</sup> (**i**) and MMP7<sup>+</sup> (**j**) tumor cells in IF-stained tumors of *Stat1*<sup>flox/flox</sup> *Apc*<sup>Min</sup> (si: 7 tumors, 3 mice; colon: 12 tumors, 3 mice) and *Stat1*<sup>ΔIEC</sup> *Apc*<sup>Min</sup> (si: 9 tumors, 3 mice; colon: 11 tumors, 3 mice) mice. **k, l** Quantification of percentages of MMP7<sup>+</sup> IDO1<sup>+</sup> tumor cells between IDO1<sup>+</sup> (**k**) and MMP7<sup>+</sup> (**l**) tumor cells of *Stat1*<sup>flox/flox</sup> *Apc*<sup>Min</sup> tumors. **m, n** IHC staining (**m**) and quantification (**n**) of IDO1<sup>+</sup> Paneth cells in early and advanced *Stat1*<sup>flox/flox</sup> *Apc*<sup>Min</sup> tumors (si and colon: 12 early adenomas, 5 mice each and 13 advanced tumors, 4 mice each). Scale bars indicate 50 μm. si: small intestine. n.d.: not detectable. Bars represent mean ± SEM.

cells in immunocompetent C57BL/6 hosts (Fig. 4b, c), which were strongly infiltrated with CD3<sup>+</sup> T cells (MC38<sup>ΔIdo1-GFP-2</sup> cells are shown in Fig. 3d, e). In contrast, growth of MC38<sup>ΔIdo1-GFP-2</sup> cells was not affected in immunocompromised NOD scid gamma (NSG) hosts, which lack mature T cells, B cells, and NK cells (Fig. 4f, g). We performed transplantation experiments with mixtures of cells to evaluate protective effects acting in *trans*. MC38<sup>ΔIdo1-GFP-6</sup> cells were additionally labeled with dsRed to discriminate them from Ido1-proficient MC38<sup>wt-GFP</sup> cells. A 1 : 1 mixture of MC38<sup>wt-GFP</sup>/MC38<sup>ΔIdo1-G/RFP-6</sup> cells showed comparable growth to MC38<sup>wt-GFP</sup> cells in immunocompetent C57BL/6 hosts, indicating that MC38<sup>wt-GFP</sup> cells restored growth

of MC38<sup>ΔIdo1-G/RFP-6</sup> cells in *trans* (Fig. 4h). IHC staining detected dsRed-positive cells close to the expected percentage in mixed tumors (Fig. 4i, j). Moreover, the prominent CD3<sup>+</sup> T-cell infiltration in MC38<sup>ΔIdo1-GFP</sup> tumors (Fig. 4d, e) was abolished in mixed tumors (Fig. 4k). IHC staining revealed strong infiltration of Granzyme B<sup>+</sup> immune cells in MC38<sup>ΔIdo1-G/RFP-6</sup> tumors, which was also abolished in mixed tumors (Fig. 4l). These data demonstrate that Ido1<sup>+</sup> MC38 CRC cells are able to promote immune escape of transplanted tumors.

**Tumor cell-intrinsic Stat1-Ido1 favors progression of CRC.** We investigated correlations of STAT1 and IDO1 protein expression

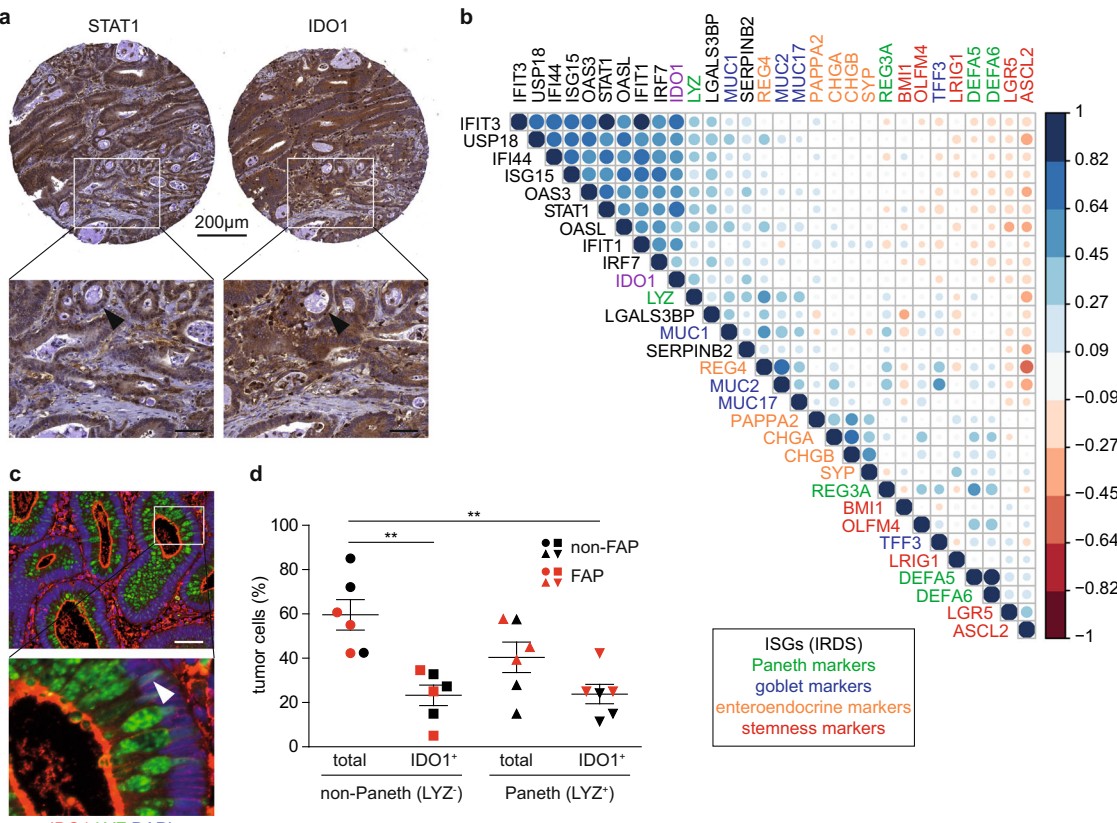

**Fig. 3 Evidence for Ido1-expressing Paneth cells in human CRC and early adenomas. a** IHC staining for STAT1 and IDO1 on consecutive tissue microarray (TMA) sections of human CRC. A tubular structure with cancer cells showing co-expression of STAT1 and IDO1 is indicated by arrowheads in the lower images (scale bars 50 μm). **b** Correlation matrix of TCGA expression data for Stat1, Ido1, IRDS genes, and markers for Paneth, Goblet, enteroendocrine, and cancer stem cells in human CRC. **c** IF staining for Lysozyme (LYZ) and IDO1 of a human adenoma biopsy of a FAP patient. The perinuclear signal for IDO1 is indicated by an arrowhead in the right high-magnification image (scale bar 50 μm). **d** IF staining was used for quantification of IDO1$^+$ non-Paneth (LYZ$^-$) and Paneth (LYZ$^+$) cells in the neoplastic epithelium of human adenomas.

in IHC-stained biopsies of 149 human T3 and T4 CRC (Supplementary Table 2). A score of 0–4 was attributed to STAT1 and IDO1 levels in tumor and stroma compartments (Supplementary Fig. 6). STAT1 or IDO1 protein in tumor or stroma cells did not correlate with overall survival and metastasis-free survival of patients (Supplementary Table 2). However, a strong correlation was observed between protein expression of STAT1 and IDO1 in tumor cells (Fig. 5a) and stroma cells (Fig. 5b). Additional analysis of TCGA data was used to increase the sample size. A correlation plot, derived from TCGA data, confirmed strong co-expression (Spearman score of 0.797) at the RNA level (Fig. 5c). Patient survival curves for Stat1 and Ido1 were similar (Fig. 5d, e), which is consistent with the co-expression of both genes. Patients with Stat1$^{high}$ or Ido1$^{high}$ cancers showed a trend towards improved survival 50 months after diagnosis (Fig. 5d, e), which might reflect beneficial effects of Stat1 in stromal immune cell activation. Consistently, immune metagene signature analysis[36] revealed strong infiltration of anti-tumor immune cells in Stat1$^{high}$ tumors (Supplementary Fig. 7a). However, they also showed increased numbers of Tregs (Supplementary Fig. 7b), which could be due to tumor cell-intrinsic Stat1-Ido1 expression. Therefore, we looked for a surrogate marker to discriminate between tumor cell-intrinsic effects of Stat1-Ido1 expression and superimposing stromal effects. We have recently identified IFN-induced Protein with Tetratricopeptide 1 (Ifit1) as a surrogate marker for Stat1 expression in breast cancer cells, whereas stroma cells were Ifit1-negative[37]. We tested whether Ifit1 can also be employed as a specific surrogate marker for Stat1-Ido1 expression

in the neoplastic epithelium of CRC. IHC staining of human biopsies demonstrated IFIT1 expression in CRC cancer cells but not in the tumor stroma (Supplementary Fig. 7c). Importantly, IFIT1 expression correlated strongly with STAT1 and IDO1 expression in CRC cancer cells (Supplementary Fig. 7c). Scoring of IDO1 and IFIT1 staining intensities (Supplementary Fig. 7d) confirmed a strong correlation of protein expression (Supplementary Fig. 7e, Pearson coefficient = 0.541). We first evaluated the prognostic value of stromal Stat1 expression in Ifit1$^{low}$ tumors. Ifit1$^{low}$ Stat1$^{low}$ and Ifit1$^{low}$ Stat1$^{high}$ tumors should display low Stat1-Ido1 expression in the neoplastic epithelium and low or high Stat1 expression in stroma cells, respectively. The Ifit1-based stratification significantly improved the prognostic value of Stat1 expression in CRC ($p = 0.03$, Fig. 5f). Patients with Ifit1$^{low}$ CRC benefited from high stromal Stat1 expression immediately after diagnosis, indicating that low expression of tumor cell-intrinsic Stat1-Ido1 sensitizes tumors to immune attack. We next mimicked conditions of our mouse models and stratified TCGA data into Ifit1$^{high}$ Stat1$^{low}$ and Ifit1$^{low}$ Stat1$^{high}$ CRC. Ifit1$^{high}$ Stat1$^{low}$ CRC (tumor cell-intrinsic Stat1-Ido1↑, stromal Stat1↓, similar to *Stat1*$^{flox/flox}$ *Apc*$^{Min}$ tumors) displayed a higher percentage of late-stage IV tumors than Ifit1$^{low}$ Stat1$^{high}$ CRC (tumor cell-intrinsic Stat1-Ido1↓, stromal Stat1↑, similar to *Stat1*$^{ΔIEC}$ *Apc*$^{Min}$ tumors) (Fig. 5g, h), suggesting that tumor cell-intrinsic Stat1-Ido1 promotes CRC progression. Moreover, we employed the DeMixT algorithm[38] to deconvolute tumor cell-intrinsic and stromal expression of Ido1 in TCGA data. We further stratified tumors into CMS1-4 consensus molecular

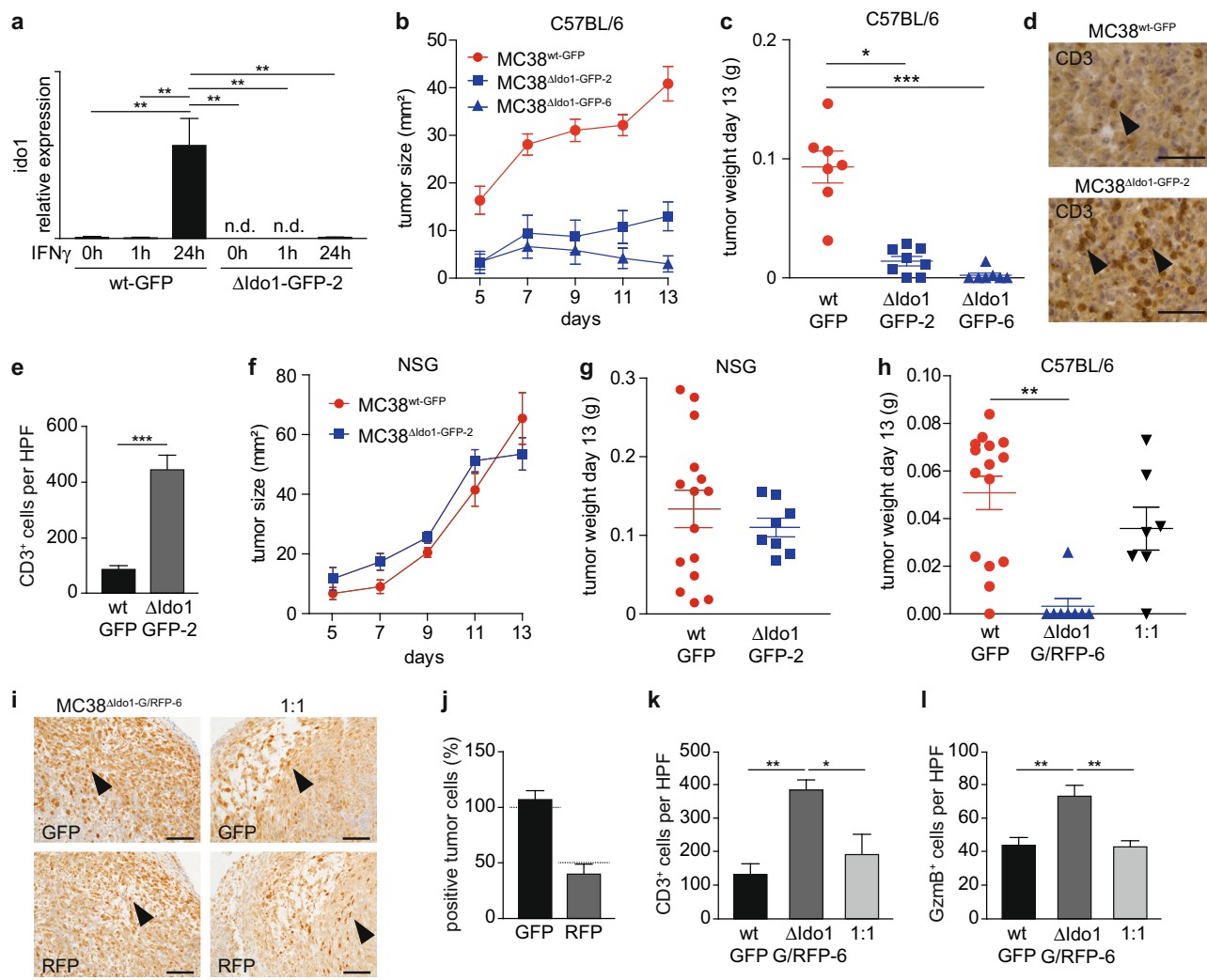

**Fig. 4 Ablation of Ido1 in MC38 cells interferes with tumor formation in immunocompetent host mice. a** qPCR for Ido1 mRNA expression in MC38[wt-GFP] and MC38[ΔIdo1-GFP-2] cells 0, 1, and 24 h after IFNγ stimulation ($n = 3$). **b, c** Tumor size (**b**) and final tumor weight (**c**) after subcutaneous injection of MC38[wt-GFP], MC38[ΔIdo1-GFP-2], and MC38[ΔIdo1-GFP-6] cells into C57BL/6 hosts (MC38[wt-GFP]: 7 tumors, 7 host mice; MC38[ΔIdo1-GFP-2]: 7 tumors, 4 host mice; MC38[ΔIdo1-GFP-6]: 7 tumors, 7 host mice). **d, e** IHC staining (**d**) and quantification (**e**) of CD3[+] infiltrating cells in tumors of MC38[wt-GFP] (7 tumors) and MC38[ΔIdo1-GFP-2] (6 tumors) cells. **f, g** Tumor size (**f**) and final tumor weight (**g**) of MC38[wt-GFP] and MC38[ΔIdo1-GFP-2] tumors in NSG hosts (MC38[wt-GFP]: 15 tumors, 15 host mice; MC38[ΔIdo1-GFP-2]: 8 tumors, 8 host mice). **h** Tumor weight of MC38[wt-GFP] (15 tumors, 15 host mice), MC38[ΔIdo1-G/RFP-6] (8 tumors, 8 host mice), and 1:1 mixed tumors (7 tumors, 7 host mice) in C57BL/6 hosts. **i, j** IHC staining (**i**) and quantitation (**j**) of GFP[+] and dsRed(RFP)[+] tumor cells in mixed tumors ($n = 3$). Expected percentages of positive cells are indicated by dashed lines. Scale bars indicate 100 μm. **k, l** Quantification of CD3[+] (**k**) and Granzyme B[+] (**l**) immune cells in MC38[GFP-wt] (three tumors each), MC38[ΔIdo1-G/RFP-6] (four tumors each), and mixed tumors (three tumors each). n.d.: not detectable. Bars represent mean ± SEM.

subtypes[39] and reinvestigated immune cell marker expression. Similarly, immune cell markers were predominantly expressed in the stroma. Moreover, the strongest expression was detected in the stroma of CMS1, a subtype characterized by immune cell infiltration[39], as exemplified for T-cell markers (Supplementary Fig. 8a, b). Without tumor stroma deconvolution, patients with Ido1[high] tumors showed a slight trend towards better prognosis (Fig. 5d, $p = 0.55$). After deconvolution, survival curves for stroma and tumor compartments were laterally reversed (Supplementary Fig. 8c). Patients with strong Ido1 expression in the neoplastic epithelial cells showed a strong trend ($p = 0.074$) towards bad prognosis. Taken together, these data suggest that tumor cell-intrinsic Stat1-Ido1 expression favors immune escape and progression of human CRC.

**IDO1[+] Paneth cells are present in normal crypts.** Similar to tumors, Ido1 mRNA expression was downregulated in small

intestinal and colonic IEC preparations of Stat1[ΔIEC] Apc[Min] mice (Fig. 6a). Therefore, we wondered whether IDO1[+] Paneth cells are present in normal crypts. IHC and ISH analyses revealed IDO1[+] vesicle-bearing Paneth cells in a subset of crypts of Stat1[flox/flox] Apc[Min] mice (Fig. 6b), which were abolished in Stat1[ΔIEC] Apc[Min] crypts (Fig. 6b, c). We detected up to three IDO1[+] cells in crypts of the small intestine and up to seven cells in colonic crypts (Fig. 6e). IDO1[+] Paneth cells were also identified in Stat1[flox/flox] mice (Fig. 6c), demonstrating that their formation does not depend on the Apc[Min] allele. They were more abundant in the distal small intestine, which has a higher bacterial load, than in the proximal small intestine (Fig. 6c). Moreover, their abundance was decreased in mice that were housed in an extra clean special pathogen-free (SPF) facility and treated with antibiotics (Fig. 6c). Treated mice also displayed reduced numbers of Lysozyme[+] Paneth cells in the proximal

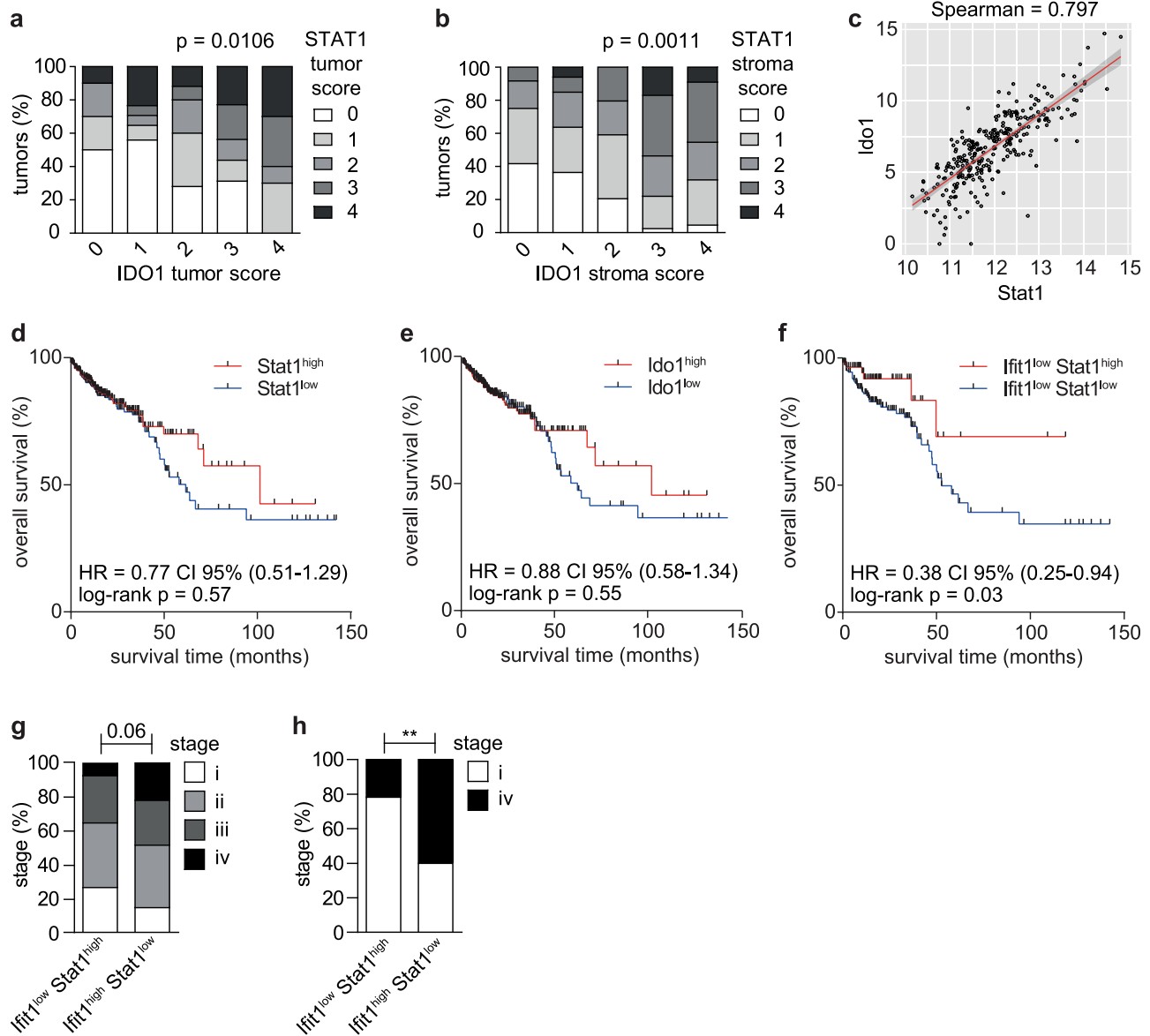

**Fig. 5 Use of Ifit1 as surrogate marker reveals a negative prognostic value of Stat1-Ido1 expression in the neoplastic epithelium of CRC. a**, **b** One hundred and forty-nine IHC-stained sections of human CRC were used to assess the correlation between STAT1 and IDO1 protein expression in the tumor cell compartment (**a**) and the stroma (**b**). **c** Scatter plot of TCGA expression data showing a strong correlation between Stat1 and Ido1 mRNA expression in human CRC. **d**–**f** Patient survival curves for Stat1high/low (**d**), Ido1high/low (**e**), and Ifitlow Stat1high/low (**f**) CRC using TCGA data. The median expression was used for stratification. **g**, **h** Stratification of human CRC into Ifit1low Stat1high and Ifit1high Stat1low cancer subtypes, and correlation with the tumor stage using TCGA data.

small intestine (Fig. 6d). These data suggest that the formation of IDO1+ Paneth cells is induced by the bacterial microbiome.

It was previously shown that the TLR9 agonist ISS DNA can induce Ido1 in intestinal epithelial cells, which protects from colitis[40]. Therefore, we isolated intestinal organoids and stimulated them with immunostimulatory (ISS) DNA (ODN 1668). Lysozyme staining showed that Paneth cells were present in organoids but ISS DNA failed to induce Ido1 (Supplementary Fig. 8d, e). However, bacteria also promote IFNγ production by CD4+ T cells in the intestinal lamina propria[41]. IFNγ readily induced Ido1 in all epithelial cells of organoids at the RNA and protein level (Fig. 6f, g and Supplementary Fig. 8e). To further investigate the role IFNγ in vivo, we performed IHC for IDO1 in intestines of $Ifngr1^{-/-}$ mice[42], which lack functional IFNγ signaling. The number of IDO1+ crypt cells was reduced in different intestinal parts of $Ifngr1^{-/-}$ mice (Fig. 6h, j).

Interestingly, $Ifngr1^{-/-}$ mice displayed reduced numbers of Lysozyme+ Paneth cells in the proximal small intestine (Fig. 6i) similar to mice with antibiotic treatment (Fig. 6d). These data demonstrate a contribution of IFNγ in induction of IDO1+ Paneth cells.

Our data suggest that a bacteria/IFNγ axis is responsible for Ido1 induction in Paneth cells. Therefore, we analyzed single-cell RNA-seq (scRNA-seq) data of Haber et al.[43] to investigate the impact of bacterial infection on Ido1 induction in Paneth cells. Paneth cell clusters were identified in t-distributed stochastic neighbor embedding (t-SNE) maps using marker genes (Supplementary Fig. 9a, b). Only two Ido1+ cells were found in t-SNE maps of healthy mice. Interestingly, they were both allocated to the Paneth-1 cell cluster, which is located in the distal small intestine and expresses the marker GM21002[43]. Haber et al.[43] derived also scRNA-seq data from bacteria- and helminth-

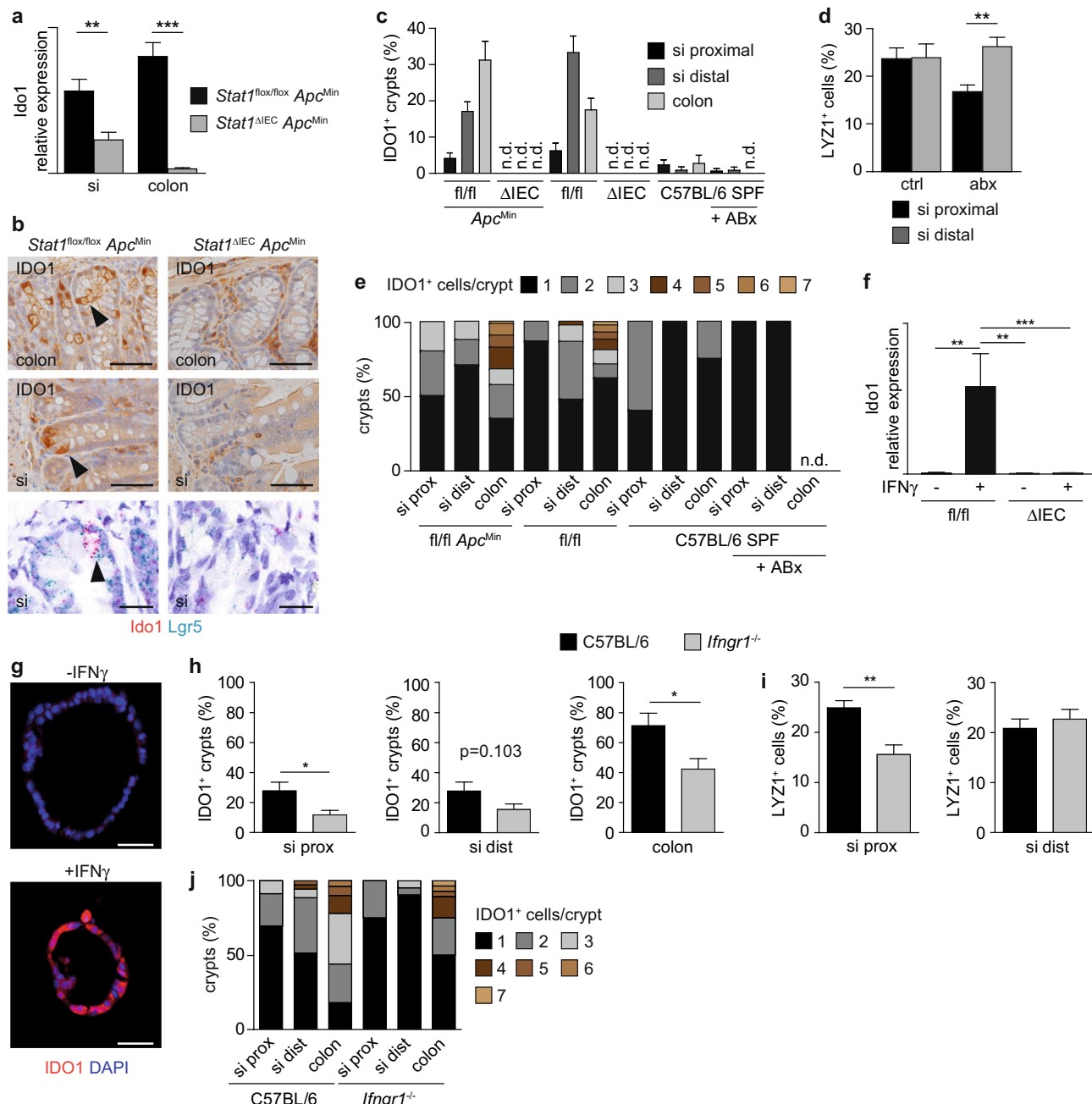

**Fig. 6 IDO1$^+$ Paneth cells are present in intestinal crypts and reduced in _Ifngr1_$^{-/-}$ mice. a** qPCR for Ido1 mRNA expression in isolated intestinal epithelial cells of _Stat1_$^{flox/flox}$ _Apc_$^{Min}$ (si: $n = 4$; colon: $n = 5$) and _Stat1_$^{\Delta IEC}$ _Apc_$^{Min}$ (si: $n = 6$; colon: $n = 5$) mice. **b** IHC staining for IDO1 (upper images) and double ISH for Ido1 and Lgr5 (lower images) in the small intestine of _Stat1_$^{flox/flox}$ _Apc_$^{Min}$ and _Stat1_$^{\Delta IEC}$ _Apc_$^{Min}$ mice. IDO1$^+$ Paneth cells are indicated by arrowheads. Scale bars indicate 20 µm. **c–e** Quantification of IDO1$^+$ crypts (**c**), LYZ1$^+$ Paneth cells/crypt (**d**), and IDO1$^+$ cells/crypt (**e**) in different intestinal compartments of co-housed _Stat1_$^{flox/flox}$ _Apc_$^{Min}$ ($n = 4$), _Stat1_$^{\Delta IEC}$ _Apc_$^{Min}$ ($n = 5$), _Stat1_$^{flox/flox}$ ($n = 4$), _Stat1_$^{\Delta IEC}$ ($n = 4$), as well as C57BL/6 mice kept at an extra clean SPF facility with and without ABx treatment ($n = 3$ each). **f** qPCR for Ido1 mRNA expression after IFNγ stimulation in intestinal organoids of _Stat1_$^{flox/flox}$ (three technical replicates, two mice) and _Stat1_$^{\Delta IEC}$ (three technical replicates, three mice) mice. **g** IF showing the induction of IDO1 upon IFNγ stimulation in tumor organoids of _Stat1_$^{flox/flox}$ _Apc_$^{Min}$ mice. Scale bars indicate 50 µm. **h–j** Quantification of IDO1$^+$ crypts (**h**), LYZ1$^+$ Paneth cells/crypt (**i**), and IDO1$^+$ cells/crypt (**j**) in different intestinal compartments of co-housed C57BL/6 ($n = 3$) and _Ifngr1_$^{-/-}$ mice ($n = 3$). n.d.: not detectable; si: small intestine; SPF: special pathogen free; ABx: antibiotics. Bars represent mean ± SEM.

infected mice. Analysis of these data revealed a prominent induction of Ido1 in Paneth cells of bacteria- but not helminth-infected mice (Fig. 7a, b, d, f). Stat1 was induced in both infection models but more prominently by bacteria (Fig. 7a–c, e). These data demonstrate that the formation of Ido1$^+$ Paneth cells is induced by the bacterial microbiome.

## Discussion

We identified an immune escape mechanism of CRC that is based on Stat1-dependent expression of Ido1 in Paneth cells. Paneth cell markers have previously been linked with intestinal tumorigenesis but the significance of the observations remained unclear. The markers Pla2g2a and Mmp7 were identified as modifiers of Min[44] and loss of Mmp7, which is essential for Paneth cell

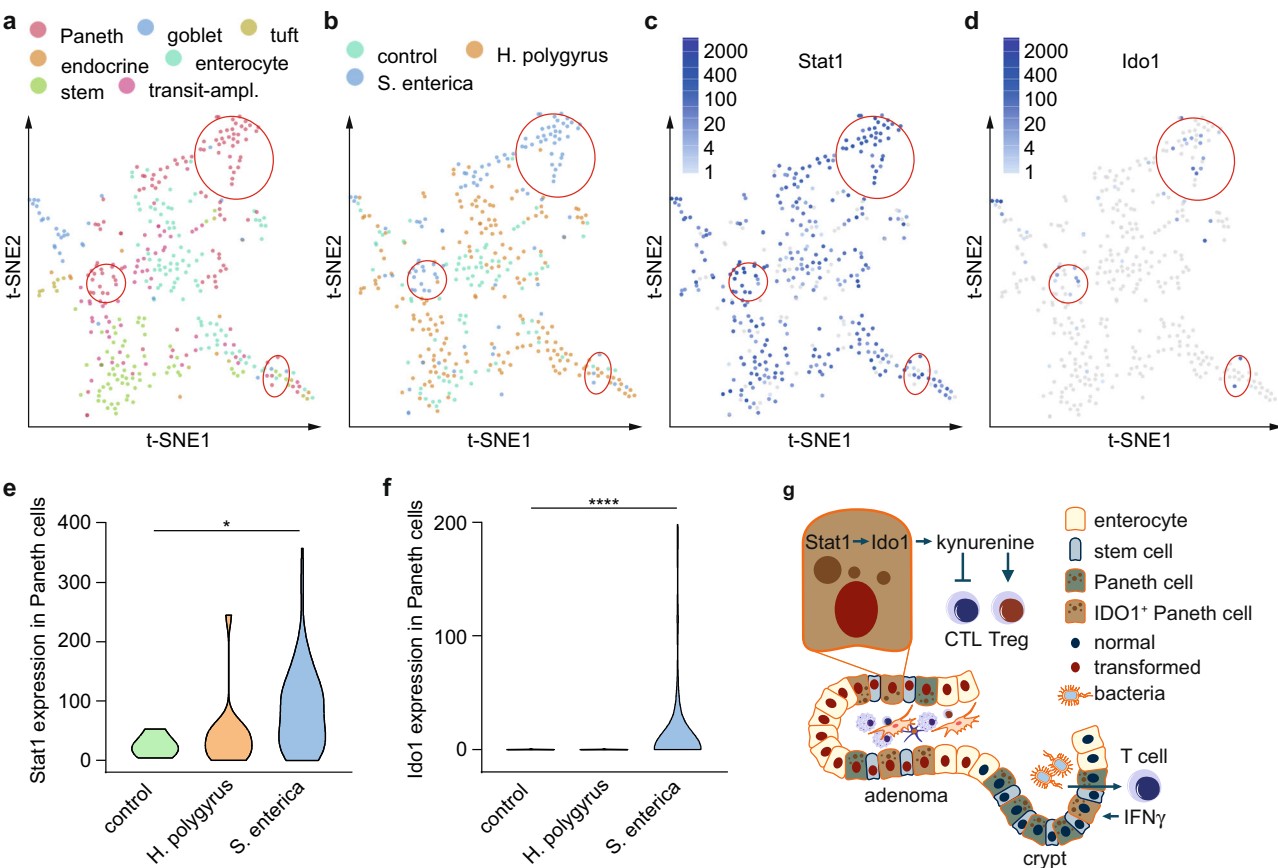

**Fig. 7 Expression of Ido1 and Stat1 is induced in Paneth cells following bacterial infection. a–d** t-SNE maps obtained from single-cell RNA-seq (full-length sequencing) of 389 epithelial cells from the small intestine of 7–10-week-old C57BL/6J mice without infection (control) and with infection (*Heligmosomoides polygyrus* or *Salmonella enterica*). Three paneth cell-enriched areas, which show increased expression of Ido1 following infection with *Salmonella enterica* are highlighted (red circles). **e, f** Violin plots for the expression of Stat1 and Ido1 (transcripts per million) in Paneth cells with and without infection. Single-cell RNA-seq data were generated by Haber et al.[43] (GEO database: GSE92332). **g** Model how IDO1+ Paneth cells promote immune escape of CRC (for details, see discussion). CTL: cytotoxic T lymphocyte; Treg: regulatory T cell.

function[45], interfered with $Apc^{Min}$-induced tumor formation[46]. Moreover, expression of Paneth markers correlated with increased risk for dietary-induced sporadic intestinal cancer in mice[47] and a Paneth cell-associated gene expression pattern was identified in human intestinal tumors[48]. The presence of IDO1+ Paneth cells in intestinal cancers might provide an explanation for these observations. IDO1 increases local kynurenine levels and depletes tryptophan. Effector T cells respond to tryptophan depletion with cell cycle arrest[49] and kynurenine promotes Treg differentiation via the aryl hydrocarbon receptor AhR[29]. This promotes an immune-tolerant microenvironment with reduced CD8+ T-cell activities and expansion of Tregs[50–52]. Ido1 over-expression is commonly observed in human CRC and associated with reduced serum tryptophan levels, whereas kynurenine metabolites are increased[53–55]. Localization studies have shown that Ido1 is expressed by infiltrating myeloid cells and neoplastic epithelial cells[56–58], and both cellular compartments could contribute to kynurenine production. Our results suggest that the neoplastic epithelium is an important source for kynurenine, because loss of IDO1+ Paneth cells in $Stat1^{\Delta IEC} Apc^{Min}$ tumors resulted in significantly reduced kynurenine levels that were not compensated by stromal kynurenine production. We speculate that neoplastic cells are major producers of kynurenine in tumors, whereas stromal myeloid cells use different metabolic routes. A contribution of Ido1-expressing neoplastic epithelial cells to immune escape has also been found in pancreatic ductal adeno-carcinomas[59] and high Ido1 expression in neoplastic epithelial cells at the invasive front is an independent adverse prognostic factor for overall survival and metastasis in CRC[58,60,61].

Stat1 is considered as a tumor suppressor in solid cancers[9] and we expected tumors of increased size in $Stat1^{\Delta IEC} Apc^{Min}$ mice. However, tumors were smaller and contained reduced numbers of Tregs and increased numbers of CD8+ T cells. Similar neo-plastic and immunologic aberrations were observed in $Ido1^{-/-} Apc^{Min}$ tumors[62]. This indicates that loss of IDO1+ Paneth cells and corresponding immunological consequences surpassed tumor-promoting effects of Stat1 deletion in $Stat1^{\Delta IEC} Apc^{Min}$ tumors. Interestingly, tumor formation was not affected in $Stat1^{-/-} Apc^{Min}$ mice[63] but this study neglected compensating effects of stromal Stat1 deletion, which interferes with immuno-surveillance and alleviates the need for immunosuppression.

Most patients develop sporadic CRC, whereas colitis-associated CRC (CAC) affects only 1–2% of human cases. Recent studies demonstrated that specific deletion of Ido1 in intestinal epithelial cells interfered with AOM-DSS-induced CAC formation in mice[30]. The oncogenic function of Ido1 in CAC was attributed to tumor cell-intrinsic phosphatidylinositol-3-kinase–Akt-mediated nuclear translocation of β-catenin rather than immunosuppression[30]. Our results showed that Stat1 ablation and corresponding loss of IDO1+ tumor cells did not affect nuclear β-catenin levels in sporadic $Apc^{Min}$ tumors. This suggests that Ido1 promotes the formation of sporadic CRC and CAC through different mechanisms. Moreover, the neoplastic and immunological con-sequences of epithelial Stat1 deletion in CAC are different from

sporadic CRC. We found increased AOM-DSS-induced CAC formation in male Stat1$^{\Delta IEC}$ mice[64]. Stat1$^{\Delta IEC}$ tumors contained reduced numbers of CD8$^+$ T cells[64], although IDO1$^+$ Paneth cells were absent (unpublished data). Therefore, Stat1-dependent IDO1$^+$ Paneth cells might be particularly important for the development of sporadic tumors but dispensable for CAC.

It is challenging to deduce prognostic information of tumor cell-intrinsic Stat1 and Ido1 expression from CRC TCGA data, because both genes are expressed in neoplastic cells and immune cells. Correspondingly, good prognosis of CRC patients with Stat1$^{high}$ tumors might be primarily caused by enhanced anti-tumor activity of Stat1$^{high}$ immune cells[7], whereas prognostic information of Stat1 expression in neoplastic cells is masked. Using IHC staining of human samples, we identified IFIT1 as a surrogate marker for STAT1-IDO1 expression in cancer cells of human CRC. IFIT1 was not detectable in stromal cells and regulates the replication of viruses, a function that should not impact on human CRC prognosis. Ifit1 surrogate expression enabled us to discriminate between tumor cell-intrinsic and stromal Stat1-Ido1 functions in bulk gene expression TCGA data. These analyses suggested that tumor cell-intrinsic Stat1-Ido1 expression promotes progression of human CRC, correlates positively with Treg numbers and desensitizes tumors to immune attack. Moreover, Stat1-Ido1 expression correlated with Lysozyme expression in human TCGA data and IDO1$^+$ Paneth cells were present in adenomas of FAP patients, indicating that human CRC also contain neoplastic IDO1$^+$ Paneth cells.

Stat1-dependent IDO1$^+$ Paneth cells were also found in normal murine crypts. They did not depend on Apc$^{Min}$ but the presence of the Min mutation affected their spatial distribution in the distal small intestine and colon. Extra clean SPF conditions and treatment of mice with antibiotics substantially reduced the number of IDO1$^+$ crypts in all parts of the intestine. Moreover, IDO1$^+$ Paneth cells in normal crypts were enriched in the distal small intestine, which has a high bacterial load, and we could identify Ido1 induction in Paneth cells of bacteria-infected mice using scRNA-seq data. This suggests that IDO1$^+$ Paneth cells are induced by the local microbiome. TLR9 and IFN signaling are candidate pathways that could promote Ido1 expression in Paneth cells. The tumor studies with Ifnar1$^{\Delta IEC}$ mice and ISS DNA-treated organoids suggest that Ido1 is not induced by type I IFN or TLR9 signaling. In contrast, IFNγ readily induced Ido1 in epithelial cells of organoids and IDO1$^+$ Paneth cells were reduced in the intestine of Ifngr1$^{-/-}$ mice. However, in contrast to Stat1$^{\Delta IEC}$ mice, IDO1$^+$ Paneth cells were not completely abolished in Ifngr1$^{-/-}$ mice, indicating that additional factors are implicated in Stat1-Ido1 induction. Ido1 is an IFNγ-inducible gene in human and murine tumor cell lines[33]. The microbiome induces IFNγ production by mucosal T cells in mice[41] and in humans[65], and depletion of bacteria reduces IFNγ levels[41]. This indicates that interaction of lamina propria cells with the microbiome leads to the production of type II IFN that induces Stat1-dependent Ido1 expression in Paneth cells of distinct intestinal crypts.

In summary, we identified Stat1-dependent IDO1$^+$ Paneth cells in intestinal tumors and normal intestinal crypts. They might represent bone fide Paneth cells but need Stat1 for Ido1 expression. IDO1$^+$ Paneth cells could act as local immunosuppressors to prevent aberrant immune cell activation in response to bacteria. Hence, they could also provide immune-privileged niches for tumor formation (Fig. 7g). Early adenomas might use these niches to shield anti-tumor immune attack during elimination and equilibrium phases of immunoediting. Consistent with this idea, neoplastic IDO1$^+$ Paneth cells were particularly abundant in early adenomas of Apc$^{Min}$ mice. Of note, tumor formation was impaired in Apc$^{Min}$ mice kept under germ-free conditions[66]. Targeting IDO1$^+$ Paneth cells might improve efficacy of immunotherapy in microsatellite-stable CRC patients. Besides representing a conceptual advance, our findings will improve precision oncology of CRC.

## Methods

**Mice**. Mice with floxed alleles of Stat1[24] or Ifnar1[34] were crossed to Villin-cre mice[23]. Villin-cre Stat1$^{flox/flox}$ and Villin-cre Ifnar1$^{flox/flox}$ animals were crossed with Apc$^{Min}$ mice[22] (Jackson Laboratory) to generate Villin-cre Stat1$^{flox/flox}$ Apc$^{Min/+}$ (Stat1$^{\Delta IEC}$ Apc$^{Min}$) and Villin-cre Ifnar1$^{flox/flox}$ Apc$^{Min/+}$ (Ifnar1$^{\Delta IEC}$ Apc$^{Min}$) mice. Mice were kept on a C57BL/6 genetic background and housed under standard conditions at the Dezentrale Biomedizinische Einrichtung of the Medical University Vienna (Stat1$^{\Delta IEC}$ Apc$^{Min}$) and the Zentrale Versuchstieranlage of the Medical University Innsbruck (Ifnar1$^{\Delta IEC}$ Apc$^{Min}$). Experiments were performed with adult (6–8 weeks old) or tumor-bearing (4 months old) male or female mice. To deplete commensal gut microbiota, adult wild-type C57BL/6J mice were given ampicillin (1 g/l), vancomycin (0.5 g/l), neomycin sulfate (1 g/l), and metronidazole (1 g/l) in drinking water for 28 days. Water was changed every third day to ensure antibiotic stability. All mouse experiments were performed in accordance with Austrian and European laws (license numbers BMWFW-66.009/0191-WF/V/3b/2015 and BMWFW-66.009/0189-WF/V/3b/2015) and with the general regulations specified by the Good Science Practices guidelines of the Medical Universities Vienna and Innsbruck.

**Human material**. Patient material from the Austrian Breast and Colorectal Cancer Study 91 (ABCSG trial 91, NCT00309543) was used, which is a prospective, multicenter, randomized trial comparing the efficacy of adjuvant chemotherapy in stage II colon cancer[67]. All patients provided written consent and the study was approved by the ethics committees at the participating institutions.

**Isolation of intestinal epithelial cells**. Intestinal epithelial cells were isolated from 10–12-week-old mice as described previously[68].

**Histology, immunohistochemistry, and immunofluorescence**. Intestines were flushed with phosphate-buffered saline (PBS) and 4% paraformaldehyde, fixed and embedded in paraffin as swiss rolls. Swiss rolls were cut into 2 μm sections and IHC/IF-stained with standard procedures using antibodies against β-catenin (Becton Dickinson, 610153, 1 : 80), BrdU (BrdU In-Situ Detection Kit, Becton Dickinson, 550803), cleaved Caspase 3 (Cell Signaling, 9661, 1 : 200), Endomucin (eBioscience, 14-5851-82, 1 : 500), GR1 (Serotec, MCA771GA, 1 : 200), Granzyme B (Abcam, ab4059, 1 : 200), IDO1 (Biolegend, 122402, 1 : 80), iNOS (Biolegend, 610431, 1 : 200), Ki67 (Novocastra, NCL-KI67-P, 1 : 1000), Lysozyme (Dako, A009902, 1 : 100), p-STAT1 (Cell Signaling, 9167S, 1 : 100), p-STAT3 (Cell Signaling, 9145, 1 : 100), STAT1 (Santa Cruz, sc-592, 1 : 500), STAT3 (Santa Cruz, sc-7179, 1 : 80), Synaptophysin (GeneTex, GTX100865, 1 : 1000), GFP (Roche, 11814460001, 1 : 1000), red fluorescent protein (RFP) (Rockland antibodies and assays, 600-401-379S, 1 : 1000), CD3 (Neomarker RM9107, 1 : 100), MMP7 (Cell Signaling, 3801, 1 : 100). IHC staining on human samples was performed using antibodies against IFIT1 (Sigma Aldrich, HPA055380, 1 : 500), IDO1 (Biolegend, 122402, 1 : 100), p-STAT1 (Cell Signaling, 9167S, 1 : 100), and STAT1 (Cell Signaling, 14994, 1 : 1000).

**CRISPR/Cas9 of MC38 cells and transplantation**. MC38$^{\Delta Ido1-GFP}$ cells were generated using CRISPR-Cas9 as described previously[69]. Ido1 exon 6 was targeted using the following oligos: 5′-CACCTCCTGGTGGGGACTGCGACA-3′ (forward) and 5′-AAACTGTCGCAGTCCCCACCAGGA-3′ (reverse). Frequency of insertions/deletions in the transfected cell pool was estimated using the TIDE analysis software. The following primers were used for target site amplification: 5′-AACTCAGGGCTTTGAGAATGT-3′ and 5′-TTCATCCACTAAGCCACCCC-3′. Single cells derived from the initially targeted cell pool were expanded independently, sequenced for DNA modifications using the above mentioned primers, and used for transplantation experiments. To label MC38$^{\Delta Ido1-GFP}$ cells with dsRed, 10 μg of DsRed-pLenti plasmid (gift from Venugopal Bhaskara), 8 μg of packaging vector (psPAX2, Addgene plasmid # 12260), 3 μg of envelope vector (pVSV-G, Addgene plasmid # 14888), and 61 μl of 2 M CaCl$_2$ were diluted to 500 μl in ddH$_2$O. The solution was then mixed with 500 μl of 2× HBS (50 mM HEPES, 10 mM KCl, 12 mM Dextrose, 280 mM NaCl, 1.5 mM Na$_2$HPO$_4$ pH 7.05) and incubated for 10 min at room temperature, before being added to HEK293T cells for virus production. Target cells were incubated with virus containing supernatant for 5 days. Cells 10$^6$ were injected subcutaneously into the flanks of 8–9-week-old male C57BL/6J mice or NSG mice. To account for host effects, cells of different genotypes were implanted in the left and right flanks of the same mouse. Mixed MC38$^{wt-GFP}$/MC38$^{\Delta Ido1-G/RFP}$ tumors were evaluated for the presence of GFP- and RFP-positive cells via IHC. Individual MC38$^{wt-GFP}$ and MC38$^{\Delta Ido1-G/RFP}$ tumors with 100% GFP$^+$ or GFP$^+$/dsRed$^+$ cells were used for normalization of data.

**MC38 cell stimulation**. MC38$^{wt-GFP}$ and MC38$^{\Delta Ido1-GFP}$ cells were cultured in Dulbecco's modified Eagle's medium containing 10% fetal calf serum (FCS), 1%

Penicillin/Streptomycin (10,000 U/ml) and 1% L-glutamine (200 mM). At 70–80% confluency, cells were stimulated with 100 ng/μl IFNγ (Immunotools, 12343536) for 1 h and 24 h, in triplicates.

**Quantification and grading of Apc^Min tumors.** Swiss rolls were stained with hematoxylin and eosin, scanned with a Pannoramic Midi Slide Scanner (3D Histec), and histomorphometrically analyzed with Definiens™ Developer software (Definiens). Grading was performed by a board certified pathologist (L.K.).

**Flow cytometry.** Intestinal tumors from single mice were pooled, minced and digested in 2 ml PBS containing 0.25% (v/v) FCS and 0.25% (w/v) collagenase IV (Life technologies, 17104-019) for 45' at 37 °C under shaking. After straining through a 70 μm mesh and washing twice with 30 ml PBS, cells were incubated with TruStain fcX (Biolegend, 101320) and Zombie Aqua Fixable Viability Kit (Biolegend, 423102). Extracellular staining was performed using antibodies against CD8a (Biolegend, 100728), CD45 (Biolegend, 103128), CD4 (Biolegend, 100408), CD3e (eBioscience, 35-0031-82), and CD25 (eBioscience, 25-0251-81). Cells were fixed (Fixation/Permeabilization Buffer, eBioscience, 00-5123-43), permeabilized (Permeabilization Buffer, eBioscience, 00-8333-56), and intracellular staining of FOXP3 (Biolegend, 320011) and Granzyme B (Biolegend, 515405) was performed. Data were collected using a FACS Fortessa (BD) and analyzed with FlowJo software.

**Enzyme-linked immunosorbent assay.** Single tumors of the small intestine and colon were homogenized in 60 μl PBS and centrifuged. Supernatants were used for kynurenine ELISA (EMELCA Bioscience, MBS043489) according to the manufacturer's instructions. The results were normalized for the amount of protein.

**In-situ hybridization.** Duplex ISH was performed on formalin-fixed paraffin-embedded tissue samples using RNAscope 2.5 HD assay (ACD, 322436) with probes against Lgr5 (ACD, 312171) and Ido1 (ACD, 315971) according to the manufacturer's instructions.

**RNA sequencing.** Total RNA from tumors was extracted using TRIzol Reagent (Thermo Fisher Scientific, 15596018) and processed for sequencing using the TruSeq RNA Sample Preparation Kit (Illumina, Inc.) according to the manufacturer's protocol. mRNA was purified using poly(T)-oligo-attached magnetic beads, fragmented, and applied to first-strand complementary DNA (cDNA) synthesis. Second-strand cDNA synthesis was performed using DNA polymerase I and RNase H. cDNA was end-repaired, A-tailed, ligated to adapters, and amplified to create the final cDNA library for sequencing (HiSeq2000, Illumina, Inc). TopHat2 algorithm was used to align raw RNA-seq data to mm10. Aligned bam files were deposited in ArrayExpress database (E-MTAB-5083). Differentially expressed genes were identified using DeSeq2 algorithm. An adjusted $p < 0.005$ and a fold change $>2$ or $< -2$ were defined as cut-off for differentially expressed genes. GO enrichment analyses were performed using GOrilla software.

**scRNA-seq analysis.** Pre-processed droplet-based scRNA-seq datasets from Haber et al.[43] (GEO; GSE92332) were re-analyzed using the R package Seurat. For comparison of Ido1 expression in Paneth cell clusters, the infection model datasets "SH_Salmonella" and "SH_Hpoly" were used, as well as the according control sets. Different infection durations (3 days and 10 days) within the "SH_Hpoly" dataset were pooled. Furthermore, sequencing data of intestinal cells, specifically sorted with focus on large cells to improve Paneth cell yield, were analyzed. Dimensionality reduction was performed using gene expression data for a subset of variable genes. The variable genes were selected based on dispersion of binned variance to mean expression ratios using FindVariableGenes function of Seurat[70] followed by filtering of ribosomal protein and mitochondrial genes. Next, principal component analysis (PCA) was performed and the data were reduced to the top 15 PCA (infection model)/10 PCA (large cells) components (number of components was chosen based on SDs of the principal components—in a plateau region of an elbow plot). Graph-based clustering of the PCA reduced data with the Louvain Method was used after computing a shared nearest-neighbor graph[70]. The clusters were visualized on a two-dimensional map produced with t-SNE. The VlnPlot function was applied to show expression probability distributions across the clusters and the FeaturePlot function to visualize feature expression within the clusters on a t-SNE plot. These methods were performed for marker genes of our cells of interest to identify Paneth and goblet cell clusters. Violin plot expression levels are depicted on a log transcripts per million (TPM) scale per cluster. Feature plot depicts a color scale for average gene expression. To identify further clusters containing Ido1+ cells (TA/Stem, Tuft, enterocytes), the top 50 specific marker genes for each cluster identified and described in Haber et al.[43] were selected using the CaseMatch function and aggregated and matched to gene expression profiles of the clusters identified within this analysis using the MetaFeature function (calculation of relative contribution of each feature to each cell for given set of features).

**Polymerase chain reaction.** Wild-type, floxed and deleted Stat1 alleles were amplified using 5′-TAGGGCTCCCTCTTTCCCTTC-3′, 5′-

ACACCATTGGCTTCACCTTC-3′, and 5′-CCCCTGTCATCTGGAGTGAT-3′ primers. The Cre transgene was detected with 5′-CGGTCGATGCAACGAGT-GATGAGG-3′ and 5′-CCAGAGACGGAAATCCATCGCTCG-3′ primers. Apc^Min genotyping was performed using 5′-TCTCGTTCTGAGAAAGACAGAAGCT-3′ and 5′-TGATACTTCTTCCAAAGCTTTGGCTAT-3′ primers, and digestion of amplicons with HindIII.

**Quantitative PCR.** RNA was isolated with TRIzol (Life Technologies, 15596-018) and reverse transcribed with QuantiTect Reverse Transcription Kit (Qiagen, 205313). qPCR was performed using Fast SYBR Green Mastermix (Thermo Fisher Scientific, 4385616) and Applied Biosystems 7500 Fast Real Time PCR System with primers 5′-TGGTGAAATTGCAAGAGCTG-3′ and 5′-TGTGTGCGTACCCAA-GATGT-3′ for Stat1, 5′-ATGTGGGCTTTGCTCTACCA-3′ and 5′-AAGCTGCCC GTTCTCAATCA-3′ for Ido1, and 5′-TGTTTGTGATGGGTGTG-3′ and 5′-TAC TTGGCAGGTTTCTC-3′ for Gapdh.

**Statistics and reproducibility.** Sample sizes and numbers of replicates are described in detail in the figure legends. Biological replicates were defined as parallel measurements of biologically distinct samples (mice in most cases). Each experiment was repeated at least three times. All values are given as means ± SEM. Normal distribution of data was tested and appropriate tests were performed: comparisons of two groups were calculated with unpaired Student's t-test or Mann–Whitney U-test. For more than two groups, one-way analysis of variance and Bonferroni's post-hoc test or Kruskal–Wallis test, and Dunn's post-hoc test were used. For analysis of the tumor and TMA gradings, $\chi^2$-test was used. Survival analyses using clinical data from CRC TCGA patients were performed using log-rank testing and GraphPad Prism 6 software. Correlation analyses of TCGA data were calculated using cor function of R3.2.1 software and visualization was performed using corrplot and ggplot2 packages. No sample size estimation was performed. Samples were excluded as outliers according to Grubbs' test ($\alpha = 0.05$). Experiments were performed and analyzed in a blinded, randomized manner. Significant differences between experimental groups are stated as: *$p < 0.05$, **$p < 0.01$, ***$p < 0.001$, or ****$p < 0.0001$.

**Reporting summary.** Further information on research design is available in the Nature Research Reporting Summary linked to this article.

## Data availability

RNA sequencing data were deposited in ArrayExpress database, accession number E-MTAB-5083. Previously generated single-cell RNA sequencing data analyzed here can be found in GEO (GSE92332). Processed (MapSplice aligned, RSEM quantified and upper-quartile normalization standardized; Level 3 RnaSeqV2) RNA sequencing data of the COADREAD dataset were obtained from The Cancer Genome Atlas (TCGA) database. All other data that support the findings of this study are available from the corresponding author upon reasonable request.

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

## Acknowledgements

We thank Latifa Bakiri and Erwin F. Wagner for their critical reading of the manuscript. We thank Nina Braun for support with single-cell RNA-seq analysis. This work was supported by the Austrian Science Fund (FWF) Doktoratskolleg-plus grant "Inflammation and Immunity" (R.E., M.S., M.M., and B.S.), the FWF grants P29222-B28 (R.E.), F4709-B20 (G.H.), P26011 (L.K.), P32900 (E.C.), SFB F6101 (M.M. and B.S.), SFB F6106 (M.M. and B.S.), and the European Training Network MSCA-ITN-2015-ETN ALKA-TRAS No 675712 (L.K.).

## Author contributions

Conceptualization, S.P., J. Svinka, and R.E. Methodology: S.P., J. Svinka, G.T., and E.G. Investigation: S.P., J. Svinka, I.S., I.C., M.F., P.C., L.K., M.A., M.S., R.B., D.H-B., Z.T., and G.H. Writing—original draft: R.E. Writing—review and editing: S.P., J. Svinka, I.C., E.G., M.S., Z.T., G.H., and R.E. Funding acquisition: L.K., G.H., M.S., M.M., B.S., and R.E. Resources: M.F., M.T., J. Stift, H.M., E.C., M.S., M.G., S.L., J.T., M.M., B.S., T.M., and A.K. Supervision: R.E.

## Competing interests

The authors declare no competing interest.
