## [Peer Review File · Communications Biology]

Reviewers' comments:

Reviewer #1 (Remarks to the Author):

Pflugler et al use a combination of mouse models, immunohistochemistry of patient samples and bioinformatics analyses to demonstrate the roles of Stat1 and Ido1. While some of the mouse experiments presented are well-done and interesting, the manuscript as a whole is unwieldy, brings together disparate models and lines of evidence, and does not definitely demonstrate its key claims.

An intriguing observation in Fig 1j is the increase in iNOS and GzmB in Stat1-deficient small adenomas but not in larger adenomas measuring over 0.5 mm³. This suggests that the role of Stat1/Ido may be restricted to the early stages of tumor initiation, and that suppression of this axis in later adenomas might be ineffectual. If borne out, this observation may provide an explanation for the current underwhelming performance of Ido1 inhibitors such as epacadostat in clinical trials in patients with advanced cancers. However, the authors do not explore this difference further or attempt to understand its basis, limiting the impact of the study.

Major points:

1. A key novel claim of this manuscript is the Stat1-mediated production of Ido1 specifically in Paneth cells. However, this claim is based exclusively on immunostaining (Fig 2), which is of poor quality and not convincing.

- The co-immunostaining of Ido1 with lysozyme should be shown at higher magnification together with a membrane marker, and ideally an additional marker of Paneth cells as well as Stat1 to clearly demonstrate the presence of Stat1+ Ido+Paneth cells. Markers of other intestinal cells including stromal cells and other epithelial cells are also needed and the Lgr5 ISH lacks clarity. Possibly repeating the ISH with a different chromogen might provide better visual contrast with the hematoxylin counterstain. Further, are the stained tumors derived from the small intestine or the colon? Clear quantification of colocalization of Ido+Stat+ with markers of Paneth and other cell types in both the small intestine and the colon (the latter being the clinically relevant entity) should be shown.
- While the authors have attempted to use bioinformatics analysis of TCGA RNAseq data to corroborate these claims, these data are derived from whole tumors, containing a mixture of tumor and stromal cells, the latter of which are a known important source of Ido1. Correlation between Ido1 and Lyz, as shown in Fig 2h, is not evidence of co-expression in the same cells.
- In order to conclusively demonstrate the Stat1-dependent expression of Ido1 in Paneth cells, the authors should ideally use either mouse models or organoids derived from mouse models that enable lineage-specific expression of cre recombinases in either paneth cells (lyz or defensin-driven Cre), LGR5+ crypt base stem cells, or non-paneth cells (Ah-Cre) to drive the deletion of Stat1 specifically in one or other lineage and assay Ido1 expression.
- An additional alternative would be to include single cell mRNA sequencing (or reanalysis of published datasets) of tumors and/or normal intestinal epithelium demonstrating the expression of Ido1 specifically in Paneth cells.

2. While the authors demonstrate the expression of Stat1 and Ido1 in human colorectal cancer cells, they do not explore whether these cells are Paneth-like. In lines 211-212 they state "unlike mouse tumors, Ido1+ cells did not show an alternating pattern...", yet they are comparing mouse adenomas with T3 and T4 invasive adenocarcinomas. Instead, the authors should stain early stage adenomas/polyps from patients for a more appropriate comparison with APCmin adenomas. Human tumors should be costained with markers of Paneth-like cells of the colon to determine whether such cells are indeed a source of Ido1.

3. In Figure 3, the authors switch from a genetically induced mouse model of mostly small intestinal

microsatellite stable adenomas – a poorly immunogenic model, to a completely different model with MC38 – a chemically induced murine colon carcinoma with high tumor mutation burden and microsatellite instability, which has been in culture for decades and is highly immunogenic. No biological rationale for the change in model system is provided. Do MC38 cells contain Paneth-like cells? A more consistent model would be to generate organoids from the APCmin tumors and use CRISPR in organoids to generate Ido deletions and repeat the experiments performed.

4. Details of the CRISPR experiment should be provided in the results and/or methods sections – was the MC38delta Ido1-GFP line used clonally derived from a single cell (as it seems)? If so, the experiments in Figure 3 should be repeated with at least one more independent delta Ido1 clone to control for clone-dependent and off-target effects.

5. The wt/delta Ido1 experiment is interesting and demonstrates a nice non-cell autonomous effect of Ido1. Is the immunosuppression induced by the introduction of wt cells only local or is there an abscopal effect? It would be interesting to implant wt and delta-Ido1 tumors on opposite flanks of the same mouse and assay the effect on distant tumor growth.

Other points:

1. Supplementary Figure 4a shows increase in Ido2 expression in Stat1 delta IEC cells that shows a decrease in Ido1 expression. Is Ido2 able to compensate for the loss of Ido1? Given the abundant expression of Stat1 and Ido in the stroma, it is unclear why Ido from these sources is insufficient to compensate and maintain kynurenine production. The authors should comment on this.

2. Fig 2. IHC is overstained.

3. Supplementary Fig, 8d. Scores for Ido1 and Ifit staining intensity should be plotted against each other and a correlation coefficient calculated.

Pflugler et al

Reviewer #2 (Remarks to the Author):

Pflugler et al present an interesting story about Stat1 mediated regulation of Ido1 in intestinal epithelial and Paneth cells in colorectal cancer. They present a combination of in vivo mouse models and analysis of human data. They unexpectedly find that Stat1 in epithelium is tumor promoting (unexpected because of the role of Stat1 in other cancers and role of IFN γ R in epithelium) and link it to the ability of Stat3 to regulate immunosuppressive Ido1. The data is based both on APCMin model and on MC38 cell model. Overall the manuscript contains novel information, is technically sound and data is well presented and described. This manuscript is very good candidate for publication. Only few points of criticism apply:

Major:

1) Interpretation of the data that Stat1 is required for the formation/ontogenesis of Ido1+ Paneth cells is not clear. What if Stat1 simply regulates Ido1 expression but otherwise does not do anything to the biology of the cell? So the cell in the absence of Stat1 is now "Ido1 expression wannabe", is Ido1 negative but otherwise is the same? Please rephrase or clarify.

2) IFN γ seems to be one of the candidates for Stat1-Ido1 induction. Can IFN γ be blocked with neutralizing antibody in the most simplistic CRC models and Ido1 expression and presence of Paneth cells assessed?

3) Antibiotic data in the context of Fig 5 is mentioned but not presented. It would be nice to see the data on antibiotic treatment and : Paneth cell number, Ido1+ cell number, Ido1 expression, IFN γ expression and Stat1 activation.

Minor:

- 1) Introduction. Below 13% survival for metastatic cancer is only for distant/liver metastasis, not for lymph node metastasis. Please correct /clarify.
- 2) Ref #62 is incomplete. Please check all other ref's.

Reviewer #3 (Remarks to the Author):

In their manuscript "Ido1+ Paneth cells promote immune escape of colorectal cancer" Pflügler et al. show that Stat1-induced Ido1 expression in intestinal Paneth cells helps tumor cells evade immune-surveillance. I enjoyed reading this manuscript. It seems the experiments were carefully planned and executed while also being well controlled. The only major issue I see is the discrepancy between the proposed role of Stat1 and Ido1 in enabling immune-surveillance evasion in mice, and the survival curves in human patients in which high Stat1 and Ido1 expression leads to higher survival. The use of Ifit1 as a surrogate marker of Stat1 is not well-established in the field and is thus a weakness of the manuscript. However, as cancer is such a complicated disease, I do not find it surprising at all that Stat1 and Ido1 have multiple and perhaps contradicting roles in different cell types of heterogeneous tissues. Other than a few corrections to the text that I will outline below, I suggest the manuscript be accepted for publication with no further experiments needed.

- Line 76: First use of Stat1 and NK acronyms. Please write full name.
- Line 87: It is still debated whether Lgr5+ cells are the precursors for CRC.
- Line 93: First use of FAP acronym. Please write full name.
- Line 126: The data don't support this claim. Consider revising to "Stat3 and pY-Stat3 positive cells numbers were unchanged".
- These data (fig S3) seem important enough to be in the main manuscript and not the suppl.
- Line 157: Please explain to the readers what the meaning of iNOS+ cells, GzmB+ cells and why you tested for their presence.
- Figure 2C lacks proper labeling of mouse genotypes.
- Line 210: Is Ido1 expressed in Paneth cells of humans by IHC?
- Figure 3 is not clearly labeled. It is not understood from the figure or the legend what is the difference between b,c and f,g.
- Line 231: Please elaborate on mouse model when writing "immunocompromised NSG hosts"
- Line 261: Ifit1 acronym.
- The literature does not fully support the claim that Ifit1 is a specific surrogate marker for Stat1-Ido1. Could another marker be used?

Congratulations on this great work.

Shai Bel

Point to point response to the reviewers' comments

Reviewer 1:

Pfluegler et al use a combination of mouse models, immunohistochemistry of patient samples and bioinformatics analyses to demonstrate the roles of Stat1 and Ido1. While some of the mouse experiments presented are well-done and interesting, the manuscript as a whole is unwieldy, brings together disparate models and lines of evidence, and does not definitely demonstrate its key claims. An intriguing observation in Fig 1j is the increase in iNOS and GzmB in Stat1-deficient small adenomas but not in larger adenomas measuring over 0.5 mm². This suggests that the role of Stat1/Ido may be restricted to the early stages of tumor initiation, and that suppression of this axis in later adenomas might be ineffectual. If borne out, this observation may provide an explanation for the current underwhelming performance of Ido1 inhibitors such as epacadostat in clinical trials in patients with advanced cancers. However, the authors do not explore this difference further or attempt to understand its basis, limiting the impact of the study.

Major points:

1) A key novel claim of this manuscript is the Stat1-mediated production of Ido1 specifically in Paneth cells. However, this claim is based exclusively on immunostaining (Fig 2), which is of poor quality and not convincing. The co-immunostaining of Ido1 with lysozyme should be shown at higher magnification together with a membrane marker, and ideally, an additional marker of Paneth cells as well as Stat1 to clearly demonstrate the presence of Stat1⁺ Ido1⁺ Paneth cells. Markers of other intestinal cells including stromal cells and other epithelial cells are also needed and the Lgr5 ISH lacks clarity. Possibly repeating the ISH with a different chromogen might provide better visual contrast with the hematoxylin counterstain.

RE: Background issues limited our possibility for multiple IF staining with available antibodies. However, we included double IF staining for Ido1 and Mmp7 as a second marker for Paneth cells and show high magnification images of Mmp7⁺ Ido1⁺ Paneth cells as well as insets with high magnification of Lyz1⁺ Ido1⁺ Paneth cells (Figure 2c, e, f, g). Our results show that Ido1⁺ cells express also Mmp7 confirming that they are Paneth cells. Double-stained cells are clearly part of the neoplastic epithelium but not stroma cells. Moreover, we identified Ido1⁺ Paneth cells in a scRNA-seq dataset (see point 5).

The ISH was performed with commercially available probes using labels, optimized protocols and a hybridization oven of the company. It is difficult to further optimize the staining technology. The main objective of the staining is to reproduce the alternate

pattern of *Ido1* expression in neoplastic epithelial cells at the RNA level. In our opinion, this is convincingly demonstrated. We agree with the reviewer that the *Lgr5* ISH lacks clarity, which might be due to a low mRNA expression level of *Lgr5* in the tumors. The corresponding statement has been rephrased accordingly: “Co-expression of *Lgr5* and *Ido1* mRNA was barely detectable by ISH in *Stat1^{flox/flox} Apc^{Min}* tumor cells (Fig. 2a) indicating that *Ido1*⁺ cells are Paneth cells. However, the ISH signals for *Lgr5* were weak and not clearly attributable to individual cells. Therefore, double immunofluorescence (IF) with Paneth cell markers was performed”.

2) *Further, are the stained tumors derived from the small intestine or the colon? Clear quantification of colocalization of Ido⁺ Stat^t with markers of Paneth and other cell types in both the small intestine and the colon (the latter being the clinically relevant entity) should be shown.*

RE: We clarified the origin of stained tumors in the legend of Figure 2. We also quantified numbers of neoplastic *Mmp7*⁺ and *Mmp7*⁺ *Ido1*⁺ Paneth cells and included corresponding data (Figure 2h).

3) *While the authors have attempted to use bioinformatics analysis of TCGA RNAseq data to corroborate these claims, these data are derived from whole tumors, containing a mixture of tumor and stromal cells, the latter of which are a known important source of Ido1. Correlation between Ido1 and Lyz, as shown in Fig 2h, is not evidence of co-expression in the same cells.*

RE: We agree that bulk gene expression data are not suitable to demonstrate co-expression. We refer to points 5 and 6 of our response.

4) *In order to conclusively demonstrate the Stat1-dependent expression of Ido1 in Paneth cells, the authors should ideally use either mouse models or organoids derived from mouse models that enable lineage-specific expression of cre recombinases in either Paneth cells (lyz or defensin-driven Cre), LGR5⁺ crypt base stem cells, or non-paneth cells (Ah-Cre) to drive the deletion of Stat1 specifically in one or other lineage and assay Ido1 expression.*

RE: Corresponding experiments would require several additional mouse models and cannot be performed within the available period for resubmission of our revised manuscript. *Ido1*⁺ Paneth cells are lost in intestines and tumors of mice with epithelial-specific *Stat1* deletion. In our opinion, this clearly demonstrates that *Ido1* expression

in Paneth cells is Stat1-dependent and raises the question if it is justified to use additional experimental models (also with regard to the 3R rule of animal experimentation). We hope that the reviewer is satisfied with our arguments.

5) An additional alternative would be to include single cell mRNA sequencing (or reanalysis of published datasets) of tumors and/or normal intestinal epithelium demonstrating the expression of Ido1 specifically in Paneth cells.

This important suggestion led to an additional discovery that support our data. We analyzed the scRNA-seq data of Haber et al. (Haber, A. et al. A single-cell survey of the small intestinal epithelium, Nature 551, 333–339 (2017) doi:10.1038/nature24489). Paneth cell clusters were identified in t-SNE maps using marker genes (Supplementary Figure 9a, b). Because of low mRNA expression, only two Ido1⁺ cells were found in t-SNE maps of healthy mice. Despite the low number, it is interesting that they were both allocated to the newly identified Paneth-1 cell cluster (Haber et al.), which is located in the distal part of the small intestine (high bacterial load). Importantly, Haber et al. derived also scRNA-seq data from bacteria- and helminth-infected mice. Analysis of these data revealed a prominent induction of Ido1 in Paneth cells of bacteria- but not helminth-infected mice (Figure 7a, b, d, f). Stat1 was induced in both infection models but more prominently by bacteria (Figure 7a-c, e). These data demonstrate that formation of Ido1⁺ Paneth cells is induced by the bacterial microbiome, which was already claimed by us in the original manuscript and can now be shown convincingly.

6) While the authors demonstrate the expression of Stat1 and Ido1 in human colorectal cancer cells, they do not explore whether these cells are Paneth-like. In lines 211-212 they state “unlike mouse tumors, Ido1+ cells did not show an alternating pattern...”, yet they are comparing mouse adenomas with T3 and T4 invasive adenocarcinomas. Instead, the authors should stain early stage adenomas/polyps from patients for a more appropriate comparison with APCmin adenomas. Human tumors should be costained with markers of Paneth-like cells of the colon to determine whether such cells are indeed a source of Ido1.

RE: This is an important point and we performed double IF staining for Lysozyme and Ido1 on 14 early human colon polyps. 5 of these polyps were derived from FAP patients. Lysozyme-positive Paneth cells were frequent in most adenomas and some showed an alternating Paneth cell pattern in adenoma sheets. In particular FAP tumors showed a perinuclear signal for Ido1 in Paneth cells (Figure 3c). These data

show that neoplastic Lysozyme-positive cells Paneth cells of human colon polyps express Ido1.

7) In Figure 3, the authors switch from a genetically induced mouse model of mostly small intestinal microsatellite stable adenomas – a poorly immunogenic model, to a completely different model with MC38 – a chemically induced murine colon carcinoma with high tumor mutation burden and microsatellite instability, which has been in culture for decades and is highly immunogenic. No biological rationale for the change in model system is provided. Do MC38 cells contain Paneth-like cells? A more consistent model would be to generate organoids from the APC^{min} tumors and use CRISPR in organoids to generate Ido deletions and repeat the experiments performed.

RE: The rationale for the use of MC38 cells is now explained in the results of the revised manuscript: “Subcutaneous implantation of C57BL/6-derived MC38 cells into immunocompetent host mice is an approved method for evaluation of pre-clinical immunotherapy approaches³⁸. We transplanted GFP-labeled MC38 colorectal cancer cells to test if deletion of Ido1 in neoplastic cells mimics immunologic consequences of Ido1⁺ Paneth cell ablation in Stat1^{ΔIEC} Apc^{Min} tumors”.

MC38-derived tumors do not express Paneth cell markers but this should not weaken the conclusion of the MC38 transplantation experiments. Ido1-deficient MC38 cells were mainly used to demonstrate that: i) Ido1 expression in neoplastic cells is sufficient to prevent T cell infiltration into MC38-derived tumors and ii) Ido1-proficient MC38 tumor cells rescue growth and survival of Ido1-deficient MC38 tumor cells in trans.

8) Details of the CRISPR experiment should be provided in the results and/or methods sections – was the MC38delta Ido1-GFP line used clonally derived from a single cell (as it seems)? If so, the experiments in Figure 3 should be repeated with at least one more independent delta Ido1 clone to control for clone-dependent and off-target effects.

RE: Details of the CRISPR experiment are now provided in the methods. Moreover, we reproduced the data with an independent, second Ido1-deficient MC38 clone. The data are now included in Figure 4 of the revised manuscript.

9) The wt/delta Ido1 experiment is interesting and demonstrates a nice non-cell autonomous effect of Ido1. Is the immunosuppression induced by the introduction of wt cells only local or

is there an abscopal effect? It would be interesting to implant wt and delta-Ido1 tumors on opposite flanks of the same mouse and assay the effect on distant tumor growth.

RE: We usually implanted MC38^{wt-GFP} and MC38^{Ido1-GFP} on opposite flanks of the same mouse to account for host effects on tumor growth. For example, data in revised Figure 4h were generate in this way and demonstrate that there is no abscopal effect. We have included a sentence in the methods to clarify the experimental procedure.

Other points:

1. Supplementary Figure 4a shows increase in Ido2 expression in Stat1 delta IEC cells that shows a decrease in Ido1 expression. Is Ido2 able to compensate for the loss of Ido1? Given the abundant expression of Stat1 and Ido in the stroma, it is unclear why Ido from these sources is insufficient to compensate and maintain kynurenine production. The authors should comment on this.

The increase of Ido2 in Stat1^{ΔIEC} Apc^{Min} mice is only a trend but not significant. Both, Ido1 and Ido2 convert tryptophan to kynurenine but Ido2 has only 3-5% enzymatic activity of Ido1. Therefore, it is unlikely that Ido2 can compensate for loss of Ido1. The important issue of stromal Ido1 expression is discussed in the following paragraph of the revised manuscript: “Localization studies have shown that Ido1 is expressed by infiltrating myeloid cells as well as neoplastic epithelial cells and both cellular compartments could contribute to kynurenine production. Our results suggest that the neoplastic epithelium is an important source for kynurenine because loss of Ido1⁺ Paneth cells in Stat1^{ΔIEC} Apc^{Min} tumors resulted in significantly reduced kynurenine levels that were not compensated by stromal kynurenine production. We speculate that neoplastic cells are major producers of kynurenine in tumors whereas stromal myeloid cells use different metabolic routes. A contribution of Ido1-expressing neoplastic epithelial cells to immune escape has also been found in pancreatic ductal adenocarcinomas (PDACs) and high Ido1 expression in neoplastic epithelial cells at the invasive front is an independent adverse prognostic factor for overall survival and metastasis in CRC”.

2. Fig 2. IHC is overstained.

RE: The images have been exchanged and are now shown in Figure 3a.

3. Supplementary Fig 8d. Scores for Ido1 and Ifit1 staining intensity should be plotted against each other and a correlation coefficient calculated.

The data were generated with a scoring system for Ido1 and Ifit1 expression (high, low, not detectable). Therefore, plotting the staining intensities against each other resulted in a graph that is more difficult to interpret than the provided graph. We calculated the Pearson correlation coefficient (0.541) which is now included in the revised manuscript.

Reviewer 2

Pfluegler et al present an interesting story about Stat1 mediated regulation of Ido1 in intestinal epithelial and Paneth cells in colorectal cancer. They present a combination of in vivo mouse models and analysis of human data. They unexpectedly find that Stat1 in epithelium is tumor promoting (unexpected because of the role of Stat1 in other cancers and role of IFN γ in epithelium) and link it to the ability of Stat3 to regulate immunosuppressive Ido1. The data is based both on APCMin model and on MC38 cell model. Overall the manuscript contains novel information, is technically sound and data is well presented and described. This manuscript is very good candidate for publication. Only few points of criticism apply:

Major:

1) Interpretation of the data that Stat1 is required for the formation/ontogenesis of Ido1⁺ Paneth cells is not clear. What if Stat1 simply regulates Ido1 expression but otherwise does not do anything to the biology of the cell? So the cell in the absence of Stat1 is now "Ido1 expression wannabe", is Ido1 negative but otherwise is the same? Please rephrase or clarify.

RE: In addition to Lysozyme/Ido1 staining, double IF staining of tumors with Mmp7/Ido1 was included in the revised manuscript. Ido1-positive neoplastic tumors cells were also positive for the Paneth marker Mmp7 (Figure 2e). Although this is no final proof, it is likely that Ido1⁺ cells are bona fide Paneth cells expressing both Paneth markers. This issue is now addressed in the last paragraph of the discussion: "In summary, we identified Stat1-dependent Ido1⁺ Paneth cells in intestinal tumors and normal intestinal crypts. They might represent bone fide Paneth cells but need Stat1 for Ido1 expression".

2) IFN γ seems to be one of the candidates for Stat1-Ido1 induction. Can IFN γ be blocked with neutralizing antibody in the most simplistic CRC models and Ido1 expression and presence of Paneth cells assessed?

RE: It might be difficult to block local IFN γ in tumors. Therefore, we address this important issue in healthy intestines of IFN γ 1^{-/-} mice, which lack functional IFN γ signaling. We can show that the number of Ido1⁺ Paneth cells is reduced in different parts of IFN γ 1^{-/-} intestines. These data demonstrate that IFN γ is indeed a candidate for Stat1-Ido1 induction in Paneth cells. The data are now shown as Figure 6h-j of the revised manuscript and are discussed accordingly: "The tumor studies with Ifnar1 ^{Δ IEC} mice and ISS DNA-treated organoids suggest that Ido1 is not induced by type I IFN or

TLR9 signaling. In contrast, IFN γ readily induced Ido1 in epithelial cells of organoids and Ido1⁺ Paneth cells were reduced in the intestine of IFNGR1^{-/-} mice. However, in contrast to Stat1 ^{Δ IEC} mice, Ido1⁺ Paneth cells were not completely abolished in IFNGR1^{-/-} mice indicating that additional factors are implicated in Stat1-Ido1 induction.”

3) Antibiotic data in the context of Fig 5 is mentioned but not presented. It would be nice to see the data on antibiotic treatment and: Paneth cell number, Ido1+ cell number, Ido1 expression, IFN γ expression and Stat1 activation.

RE: Antibiotic data are shown in revised Figure 6c. Data on Paneth cell numbers are included as Figure 6d. With available antibodies, we were unable to identify Stat1 activation in intestinal epithelial cells of untreated mice. However, we provide analysis of scRNA-seq data in Supplementary Figure 9 and Figure 7 that show induction of Stat1 and Ido1 expression in Paneth cells after bacterial infection with Salmonella enterica. These data clearly indicate that Ido1⁺ Paneth cells can be induced by certain bacteria.

Minor:

1) Introduction. Below 13% survival for metastatic cancer is only for distant/liver metastasis, not for lymph node metastasis. Please correct /clarify.

RE: The statement was clarified: “Colorectal cancer (CRC) is the third most common cancer worldwide and patients with metastases in distant organs have a 5-year survival rate below 13%”.

2) Ref #62 is incomplete. Please check all other ref's.

RE: All references were checked.

Reviewer 3

*In their manuscript “*Ido1*⁺ Paneth cells promote immune escape of colorectal cancer” Pflügler et al. show that *Stat1*-induced *Ido1* expression in intestinal Paneth cells helps tumor cells evade immune-surveillance. I enjoyed reading this manuscript. It seems the experiments were carefully planned and executed while also being well controlled. The only major issue I see is the discrepancy between the proposed role of *Stat1* and *Ido1* in enabling immune-surveillance evasion in mice, and the survival curves in human patients in which high *Stat1* and *Ido1* expression leads to higher survival. The use of *Ifit1* as a surrogate marker of *Stat1* is not well-established in the field and is thus a weakness of the manuscript. However, as cancer is such a complicated disease, I do not find it surprising at all that *Stat1* and *Ido1* have multiple and perhaps contradicting roles in different cell types of heterogeneous tissues. Other than a few corrections to the text that I will outline below, I suggest the manuscript be accepted for publication with no further experiments needed.*

1) *Line 76: First use of *Stat1* and NK acronyms. Please write full name.*

RE: Full names have been included.

2) *Line 87: It is still debated whether *Lgr5*⁺ cells are the precursors for CRC.*

RE: The reviewer alludes to publications of Schwitalla, de Sousa e Melo and others. A corresponding statement and references have been included: “*Lgr5*⁺ stem cells at the bottom of intestinal crypts have been identified as possible precursor cells for CRC. However, non-stem cells can also acquire tumor-initiating capacity and *Lgr5*⁺ cancer stem cells are not essential for growth of primary tumors.

3) *Line 93: First use of FAP acronym. Please write full name.*

RE: The full name has been included.

4) *Line 126: The data don't support this claim. Consider revising to “*Stat3* and pY-*Stat3* positive cells numbers were unchanged”.*

RE: The statement was changed accordingly: “Numbers of *Stat3*- and activated pY-*Stat3*-positive cells were not changed in *Stat1*^{ΔIEC} *Apc*^{Min} tumors (Suppl. Fig. 2c-e)”.

5) *These data (fig S3) seem important enough to be in the main manuscript and not the suppl.*

RE: The data were moved to Figure 1 of the revised manuscript.

6) *Line 157: Please explain to the readers what the meaning of iNOS⁺ cells, GzmB⁺ cells and why you tested for their presence.*

RE: The meaning is now explained in the manuscript: “We analyzed expression of inducible nitric oxide synthase (iNOS) and the serine protease Granzyme B because they are markers for activation of several immune cells such as macrophages, mature dendritic cells, cytotoxic T cells or NK cells”.

7) *Figure 2C lacks proper labeling of mouse genotypes.*

RE: The labels were included.

8) *Line 210: Is Ido1 expressed in Paneth cells of humans by IHC?*

RE: We provide IF staining of human adenomas in the revised manuscript that shows Ido1 expression in neoplastic Paneth cells (Figure 3c).

9) *Figure 3 is not clearly labeled. It is not understood from the figure or the legend what is the difference between b,c and f,g.*

RE: Figure 3 is now shown as Figure 4 in the revised manuscript. The difference are the host strains, which are mentioned in the Figure legend. For clarification, we included names of host strains in revised Figures 4b, c and f, g. To be consistent, we also included the name of the host strain in revised Figure 4h.

10) *Line 231: Please elaborate on mouse model when writing “immunocompromised NSG hosts”.*

RE: The description was elaborated: “In contrast, growth of MC38^{ΔIdo1-GFP-2} cells was not affected in immunocompromised NOD scid gamma (NSG) hosts, which lack mature T cells, B cells and NK cells (Fig. 4f, g)”.

11) *Line 261: Ifit1 acronym.*

RE: The full name has been included.

12) *The literature does not fully support the claim that Ifit1 is a specific surrogate marker for Stat1-Ido1. Could another marker be used?*

RE: There is no other known marker. We have shown that Ifit1 is a marker for Stat1 expression in neoplastic epithelial cells of human breast tumors (Tymoszuk et al., BMC Cancer, 2014, 14:257. doi: 10.1186/1471-2407-14-257) and our IHC staining suggests that this is also true for human CRC (Supplementary Figure 6c, d). We could not detect Ifit1 in the tumor stroma of CRC (Supplementary Figure 6c). Moreover, we would like to draw the attention of the reviewer to Supplementary Table 1. The RNA-seq data show that Ifit1 is prominently downregulated in Stat1^{ΔIEC} Apc^{Min} tumors, which lack Stat1 in neoplastic epithelial cells, demonstrating that Stat1 regulates Ifit1 expression specifically in neoplastic epithelial cells. Therefore, we conclude that Ifit1 is a reasonable surrogate marker for epithelial Stat1 expression in CRC.

Reviewers' comments:

Reviewer #1 (Remarks to the Author):

Pfluegler et al present a revised manuscript addressing critiques. The manuscript has now been strengthened and the text has certainly been improved and clarified. It is also good to see an additional clonal MC38 derivative confirming the results of the Ido deletion experiment. Here the text subcutaneous implantation... Is an "approved" method... should be changed to is an "established" method, since there is no governing body to "approve" experimental methods. The clarification of the possible role of Ido1 in early steps of tumor initiation, and the inclusion of adenoma patient samples, are all appreciated.

Overall, while the manuscript is improved, the authors still do not provide rigorous definitive evidence of their central claim, i.e. that Paneth cells are the sole source of Ido1 in early tumorigenesis, and that it is Paneth-cell specific Ido1 expression that drives the observed phenotypes. To definitively demonstrate this claim would require a genetic model of lineage-specific ablation of Ido1 in Paneth cells. The lack of Paneth cell marker expression MC38 cells, which nevertheless have a strong phenotype upon Ido1 depletion, further undermines this claim. Although now improved, the data presented are purely correlative – using dual immunostaining of Paneth cells markers and Ido1, and reanalysis of published sRNAseq data to show co-expression of Ido1 and Paneth cell markers in the same cell populations following infection. If this correlative evidence is deemed editorially sufficient for publication, then I recommend at least strengthening this with appropriate quantification:

Specifically, Figure 2h quantifies the percentage of MMP7+ cells that are also Ido+ - but in fact what is needed is a quantification of the percentage of epithelial Ido+ cells that are also MMP7+. Similarly, the percentage of Ido + cells that are Lyz + should also be quantified. Similarly, Figure 3c needs quantification and statistics showing the percentage of epithelial Ido+ cells that are also Lyz+.

Reviewer #2 (Remarks to the Author):

I think Authors answered most of mine and other reviewers comments in a satisfactory manner. The study is solid, there are also some things not clear or not answered but that should not preclude from acceptance/publication at this time point for this journal.

Point to point response to the reviewers' comments

Reviewer 1:

Pfluegler et al present a revised manuscript addressing critiques. The manuscript has now been strengthened and the text has certainly been improved and clarified. It is also good to see an additional clonal MC38 derivative confirming the results of the Ido deletion experiment. Here the text subcutaneous implantation... is an "approved" method... should be changed to is an "established" method, since there is no governing body to "approve" experimental methods. The clarification of the possible role of Ido1 in early steps of tumor initiation, and the inclusion of adenoma patient samples, are all appreciated.

RE: We replaced the term "approved" with "established".

Overall, while the manuscript is improved, the authors still do not provide rigorous definitive evidence of their central claim, i.e. that Paneth cells are the sole source of Ido1 in early tumorigenesis, and that it is Paneth-cell specific Ido1 expression that drives the observed phenotypes. To definitely demonstrate this claim would require a genetic model of lineage-specific ablation of Ido1 in Paneth cells. The lack of Paneth cell marker expression MC38 cells, which nevertheless have a strong phenotype upon Ido1 depletion, further undermines this claim. Although now improved, the data presented are purely correlative - using dual immunostaining of Paneth cells markers and Ido1, and reanalysis of published sRNAseq data to show co-expression of Ido1 and Paneth cell markers in the same cell populations following infection. If this correlative evidence is deemed editorially sufficient for publication, then I recommend at least strengthening this with appropriate quantification: Specifically, Figure 2h quantifies the percentage of MMP7⁺ cells that are also Ido⁺ - but in fact what is needed is a quantification of the percentage of epithelial Ido⁺ cells that are also MMP7⁺. Similarly, the percentage of Ido⁺ cells that are Lyz⁺ should also be quantified. Similarly, Figure 3c needs quantification and statistics showing the percentage of epithelial Ido⁺ cells that are also Lyz⁺.

RE: We included all requested graphs for the Apc^{Min} tumors (Figure 2e, f, k). They show that >80% of Ido1⁺ tumor cells are Paneth cells which is now clarified in the manuscript text: "More than 80% of Ido1⁺ tumor cells expressed Paneth markers (Fig. 2e, k) indicating that Paneth cells are the major source for Ido1 expression in the neoplastic epithelium". Moreover, we established a triple IF-staining for E-cadherin, Mmp7 and Ido1 (Figure 2h) that clearly demonstrates the epithelial origin of Ido1⁺ Mmp7⁺ cells.

Regarding the human samples in Figure 3c: we quantified Ido1⁺ tumor cells and double-positive Ido1⁺ Lyz⁺ tumor cells in the 6 human adenomas that contained Ido1⁺ Paneth cells. About 50% of Ido1⁺ tumor cells were Paneth cells demonstrating a significant contribution to Ido1 expression (Figure 3d). A corresponding statement has been included in the manuscript: “The relative contribution of Paneth cells to Ido1 expression was assessed by IF staining of the 6 adenomas harboring Ido1⁺ Paneth cells. About 50% of Ido1⁺ tumor cells were Lysozyme-positive demonstrating a significant contribution of Paneth cells to Ido1 expression in the neoplastic epithelium (Figure 3d)”.

Reviewer 2:

I think Authors answered most of mine and other reviewers comments in a satisfactory manner. The study is solid, there are also some things not clear or not answered but that should not preclude from acceptance/publication at this time point for this journal.

RE: We thank the Reviewer for his/her helpful comments.

REVIEWERS' COMMENTS:

Reviewer #1 (Remarks to the Author):

The authors have satisfactorily addressed my concerns; the new staining and quantification provided are appreciated. I think this manuscript is now acceptable for publication in Communications Biology